

# The impact of aerosols on photolysis frequencies and ozone
# production in urban Beijing during the four-year period
# 2012–2015
Wenjie Wang[1], Min Shao[1,2*], Min Hu[1], Limin Zeng[1], YushengWu[1]
1 State Joint Key Laboratory of Environmental Simulation and Pollution Control,
College of Environmental Sciences and Engineering, Peking University, Beijing
100871, China
2 Institute for Environmental and Climate Research, Jinan University, Guangzhou
511443, China
**\* Correspondence to:**
Prof. Min SHAO
College of Environmental Sciences and Engineering, Peking University, Beijing
100871, China
Tel: +86-10-62757973; Fax: +86-10-62757973
Email: mshao@pku.edu.cn



**Abstract**

24        During the period 2012-2015, the photolysis frequencies were measured at the

Peking University site (PKUERS), a representative site of urban Beijing. We present a
study of the effects of aerosols on two key photolysis frequencies, $j(O^1D)$ and $j(NO_2)$.
Both $j(O^1D)$ and $j(NO_2)$ display significant dependence on AOD with a nonlinear
negative correlation. With the increase in AOD, the slopes of photolysis frequencies
vs AOD decrease, which indicates that the capacity of aerosols to reduce the actinic
flux decreases with AOD. In addition, the slopes are equal to $4.21\text{-}6.93 \cdot 10^{-6}$ $s^{-1}$ and
$3.20 \cdot 10^{-3}$ $s^{-1}$ per AOD unit for $j(O^1D)$ and $j(NO_2)$ respectively at SZA of 60°, both of
which are larger than those observed in the Mediterranean. This indicates that the
aerosols in urban Beijing have a stronger extinction on actinic flux than absorptive
dust aerosols in the Mediterranean. Since the photolysis frequencies strongly
depended on the AOD and the solar zenith angle (SZA), we established a parametric
equation to quantitatively evaluate the effect of aerosols on photolysis frequencies in
Beijing. According to the parametric equation, aerosols lead to a decrease in $j(NO_2)$
by 24.2% and 30.4% for summer and winter, respectively, and the corresponding
decrease in $j(O^1D)$ by 27.3% and 32.6% respectively, compared to an aerosol-free
atmosphere. Based on an observation campaign in August 2012, we used the
photochemical box model to simulate the ozone production rate $(P(O_3))$. The
simulation results shows that the monthly average net ozone production rate is
reduced by up to 25% due to the light extinction of aerosols. Through further in-depth
analysis, it was found that particulate matter concentrations maintain high level under





the condition of high concentrations of ozone precursors (VOCs and NOx), which
inhibits the production of ozone to a large extent. This phenomenon implies a
negative feedback mechanism in the atmospheric environment of urban Beijing.
**1. Introduction**

Solar radiation plays an important role in atmospheric photochemistry, driving

the photolysis of many key species. The photolysis of ozone ($O_3$), gaseous nitrous
acid (HONO), and carbonyl species, which contributes to the primary production of
HOx (Volkamer et al., 2010). The photolysis of ozone produces $O^1D$, which then
reacts with $H_2O$ to form OH radicals; these radicals are the main source of OH
radicals in the troposphere, as shown by reactions R1 and R2. The strong dependence
of OH concentration on $j(O^1D)$ was found in a number of field measurements (Ehhalt
et al., 2000; Rohrer et al., 2014; Stone et al., 2012). In addition, the photolysis of $NO_2$
produces $O^3P$, and then $O^3P$ reacts with $O_2$ to produce $O_3$, as shown by reactions R3
and R4, which is the only significant source of ozone in the troposphere
(Finlayson-Pitts et al., 2000). The photolysis frequencies of R1 and R3 are $j(O^1D)$ and
$j(NO_2)$, respectively.
$$O_3 + h\nu \ (\lambda \ < \ 330 \ nm) \rightarrow O^1D + O_2 \qquad\qquad (R1)$$
$$O^1D \ + \ H_2O \rightarrow 2OH \qquad\qquad (R2)$$
$$NO_2 + h\nu \ (\lambda \ < \ 430 \ nm) \rightarrow NO \ + \ O\left(^3P\right) \qquad\qquad (R3)$$



$$O\left({}^{3}P\right) \,+\, O_2 \rightarrow O_3 \tag{R4}$$

The photolysis frequencies are calculated by the following formula:
$$j = \int_{\lambda_1}^{\lambda_2} F(\lambda)\sigma(\lambda,\,S,\,T)\varphi(\lambda,\,S,\,T)\,\mathrm{d}\lambda \tag{E1}$$

$F(\lambda)$ is the actinic flux dependent on wavelength. $\sigma(\lambda,\,S,\,T)$ is the absorption
cross section of the species that absorbs in the wavelength range $\lambda_1$-$\lambda_2$. $\varphi(\lambda,\,S,\,T)$ is
the quantum yield of the photodissociation reaction product. $\lambda$, S and T represent
wavelength, species and temperature, respectively.
The effect of aerosols on photolysis frequencies depends on the aerosol optical
properties, SZA and altitude (Liao et al., 1999). The aerosol optical depth (AOD)
characterizes the integral of the extinction coefficient of aerosols in the vertical
direction. The light extinction of aerosols includes scattering and absorption, which
have different effects on the actinic flux. Scattering aerosols can enhance the actinic
flux throughout the troposphere, while absorptive aerosols reduce the actinic flux
throughout the boundary layer (Jacobson, 1998; Dickerson et al., 1997; Castro et al.,
2001). To distinguish between these two components, single scattering albedo (SSA)
is defined as the ratio of the scattering coefficient to the total extinction coefficient. In
areas with severe aerosol pollution, aerosols can significantly affect photolysis
frequencies and ozone production. Studies in Los Angeles (Jacobson, 1998), Mexico
City (Castro et al., 2001; Raga et al., 2001; Li et al., 2011), São Paulo (de Miranda et
al., 2005), Huston (Flynn et al., 2010), Europe (Real et al., 2011) and Russia (Pere et
al., 2015) have found that aerosols reduce ozone concentration by 5-30% by
attenuating photolysis frequencies. Studies in the eastern United States have shown



that scattering aerosols increase ozone concentration by 5-60% by increasing the
photolysis frequencies (Dickerson et al., 1997; He and Carmichael, 1999). Therefore,
it is necessary to quantitatively evaluate the effect of aerosols on photolysis
frequencies for the purpose of effective ozone prevention.

Currently, the methods for quantitatively evaluating the influence of aerosols on

photolysis frequencies mainly include radiative transfer model and parameterization
method (Madronich et al., 1993). Radiative transfer model is based on an algorithm
for calculating solar radiation and photolysis frequencies (Madronich et al., 1999).
The observed data of related influential factors of the photolysis frequencies are taken
as the model's input and the photolysis frequencies simulated are compared with the
observed value to test the simulation effect. The method comprehensively considers
the influence of aerosol optical properties on the photolysis frequencies, but it does
not necessarily reflect the true quantitative relationship in the atmosphere due to
complicated environmental conditions and thus the simulated results don't necessarily
reproduce observed values well (Lefer et al., 2003; Shetter et al., 2003; Hofzumahaus
et al., 2004). For example, the simulated slope of $j(O^1D)$ vs AOD by Fast-JX
algorithm within the CHIMERE model was significantly smaller than the observed
slope, particularly for the high SZA values (Mailler et al., 2016). The parameterization
method is based on the observation data taken from a certain region and is used to
establish the parameterized relationship between the photolysis frequencies and
optical properties of aerosols (such as AOD). The method can reflect the actual
atmospheric environment conditions; it also considers less influential factors and thus





is easy to apply (Casasanta et al., 2011; Gerasopoulos et al., 2012). The disadvantage
of this method is that the established parametric equations apply only to a specific
region and cannot be extended to other regions.

With rapid economic development and urbanization in past decades, China's

atmospheric pollution has become more and more severe, characterized by high
concentrations of particulate matter and ozone. Satellite observations indicates that
both the particulate matter and the ozone of eastern China are at higher levels
compared with other locations in the globe (Verstraeten et al., 2015; Ma et al., 2014).
Levels of pollution in the Beijing–Tianjin–Hebei are even more severe (Chang et al.,
2009; Che et al., 2008; Zhang et al., 2014, Zhang et al., 2016). Therefore, it is
necessary to study the effects of aerosols on photolysis frequencies and ozone
production in the urban areas of China.

Previous model studies have shown that aerosols in China can affect ozone

production by changing the photolysis rate. Tang et al. (2004) used a sulfur
transmission–emission model (STEM) to discover that ozone concentration in
northeastern China was reduced by 0.1–0.8% in the sandstorm due to the change in
photolysis rate. Tie et al. (2005) used a global aerosol–chemical model to show that
aerosols caused $j(O^1D)$ and $j(NO_2)$ to decrease in winter by 20%-30% and 10%-30%,
respectively, and in summer by 5%-20% and 1%-10%, respectively, resulting in 2%-5%
and 2% reductions in ozone concentration in winter and summer, respectively. Li et al.
(2011) used an air quality model to estimate the changes in the photolysis rate caused
by sulfate, nitrate, ammonium, and mineral dust aerosols in the central and eastern





regions of China from June 1 to June 12, 2006. This study showed that the daily
average $j(O^1D)$ in the troposphere at the altitude of 1 km, 3 km, and 10 km from the
ground was reduced by 53.3%, 37.2%, and 20.9%, respectively, resulting in a
decrease in the ozone concentration by 5.4%, 3.8%, and 0.1% in the three layers. Lou
et al (2014) found that with aerosols, annual mean photolysis rates, $j(O^1D)$) and
$j(NO_2)$, were simulated to be reduced by 6-18% in polluted eastern China, leading to
reductions in $O_3$ of up to 0.5 ppbv in those regions in spring and summer by using the
global chemical transport model (GEOS-Chem). However, all of these studies base
their results on model simulations. Research using long-term observational data to
evaluate the effects of aerosols on photolysis frequencies and ozone production in
China has not yet been published.

Our overall goal is to quantitatively evaluate the effect of aerosols in urban

Beijing on photolysis frequencies and thus on ozone production. First, the relationship
between $PM_{2.5}$ and AOD was investigated. Second, based on long-term observations
(2012-2015) of photolysis frequencies, we discussed the impact of AOD on photolysis
frequencies ($j(O^1D)$ and $j(NO_2)$) in urban Beijing in detail. Then, the quantitative
relationship between photolysis frequencies, AOD, and SZA was acquired by the
parameterization method, which could be used to quantitatively evaluate the effect of
AOD on photolysis frequencies in Beijing. Finally, a photochemistry box model was
used to evaluate the effect of aerosols on ozone production.





**2. Methodology**
**2.1. Measurement**

From 2012 to 2015, j(O$^1$D) and j(NO$_2$) were measured continuously at PKUERS
site. The site (39.99°N, 116.31°E) is located on the sixth floor of a campus building at
the Peking University, 20 km northwest of Tiananmen Square. The height from the
ground is about 30 m. The sampling point is surrounded by classroom buildings.
Concentration level and composition of air pollutants were thought to be similar to the
downtown so as to be representative for the whole of Beijing (Wang et al., 2010; Xu
et al., 2011; Zhang et al., 2012; Zhang et al., 2014).
The actinic flux was measured using a spectroradiometer and the photolysis
frequencies were calculated from the absorption cross section and quantum yield of
each species (Shetter and Muler, 1999). The spectroradiometer consisted of a single
monochromator with a fixed grating (CARL ZEISS), an entrance optic with a 2π
steradian (sr) solid angle quartz diffusor and a flexible optical quartz fiber bundle
connecting both components. The spectral measurements were performed with a
wavelength resolution of 2 nm, covering a wavelength range of 290-650 nm
(Hofzumahaus et al., 1999). The measured spectra were corrected for dark signal and
stray light. Descriptions of the calibration procedure and calculation of photolysis
frequencies are given in Bohn et al.(2008). The calculated photolysis frequencies had
a time resolution of 10 s and an uncertainty of ±10%.





The optical properties of aerosols were measured by a CIMEL solar photometer
(AERONET level 2 data collection, http://aeronet.gsfc.nasa.gov/) and the site selected
was the Beijing-CAMS site (39.933°N, 116.317°E), which is close to the PKUERS
site. The CIMEL solar photometer is an automatic solar-sky scanning radiometer that
uses selected spectral channels. The instrumentation, data acquisition, retrieval
algorithms, and calibration procedure conform to the standards of the AERONET
global network and have been described in detail by Fotiadi et al. (2006). The solar
extinction measurement was performed every 3 minutes in the spectral range 340–
1020 nm for the calculation of AOD at wavelengths 340, 380, 440, 500, 675, 870, 970,
and 1020 nm. Under cloudless conditions, the overall uncertainty of AOD data is ± 1%
at $\lambda > 440$ nm and ± 0.02 at shorter wavelengths. In this study, AOD at the wavelength
of 380 nm was chosen for analysis. This wavelength was selected as it is more
representative of $j(NO_2)$. The SSA data were derived from a field campaign
undertaken in August 2012. The absorption and scattering coefficients were measured
with an Aethalometer (AE-31, Magee) and a Single Wavelength Integrating
Nephelometer (Aurora-1000), respectively, with a time resolution of 1 minute.
Five-minute averages of ozone column concentration, SSA, and photolysis
frequencies were analyzed in this study. The total ozone column was obtained by OMI
(Ozone Monitoring Instrument) for the year 2012-2015, using overpass data.
The analysis of the effects of aerosols on ozone production (Section 3.4) was
based on the field campaign undertaken in August 2012. The relevant contents and
methods of observation are shown in Table 1. In addition, meteorological parameters





such as temperature, humidity, and pressure were simultaneously observed at the site.
Since the time resolution of VOCs is 1 hour, all data analyzed in Section 3.4 was
processed as 1-hour average values. In this study, we focused on the effects of
aerosols on photolysis frequencies and ozone production under cloudless conditions.
**2.2 Radiative Transfer Model Description**
We use the Tropospheric Ultraviolet and Visible (TUV) radiation model provided
by Sasha Madronich (Madronich, 1993). In order to solve the radiative transfer
equation, TUV uses the discrete-ordinates algorithm (DISORT) with 16 streams and
calculate the global irradiance spectra in 0.15 nm steps and resolution. The key
aerosol optical properties including AOD, SSA and AE are input into the model to test
the effect of aerosols on photolysis frequencies.
**2.3 Photochemical box model**
The photochemical box model used in this study is based on a regional
atmospheric chemical mechanism (RACM2) described by Goliff et al. (2013). The
mechanism includes 17 stable inorganic compounds, 4 intermediate inorganic
compounds, 55 stable organic compounds, and 43 intermediate organic compounds.
Compounds not specifically treated in RACM are incorporated into species with
similar functional groups. The isoprene-related mechanism used in this model is LIM
mechanism proposed by Peeters et al. (2009). In this study, the observed $NO_2$, CO,
$SO_2$, C2–C12 NMHCs, HCHO, photolysis frequencies, temperature, pressure, and
relative humidity were used as constraints to simulate the concentrations of reactive



radicals ($RO_2$, $HO_2$, and OH), intermediate species, and associated reaction rate
constants. HONO wasn't measured during the period and was calculated according to
the concentration of $NO_2$ and the observed ratio of HONO to $NO_2$ at an urban site in
Beijing, which had a marked diurnal cycle, a maximum in the early morning (ratio
values up to ~0.05–0.08 in summer) and a decrease during daytime to values around
0.01–0.02 (Hendrick et al., 2014). The model was spun up for two days once it started
running in order to ensure that the simulation was stable. It was assumed that the
lifetime of simulated species removed by dry deposit was 24 hours. The lifetime
corresponds to the assumed deposit rate of 1.2 cm $s^{-1}$ and a well-mixed boundary
layer height of about 1 km (Lu et al., 2012). Net ozone production is equal to the
reaction rate between peroxy radicals ($RO_2$ and $HO_2$) and NO minus the loss rate of
$NO_2$ and $O_3$ as shown in E2, E3, and E4. The ozone production rate ($P(O_3)$), the ozone
loss rate ($D(O_3)$), and the net $P(O_3)$ were calculated from the simulation results.

$$P(O_3) = k_{HO2+NO}[HO_2][NO] + \sum \left( k^i_{RO2+NO} \left[ RO^i_2 \right][NO] \right) \qquad \text{(E2)}$$

$$D(O_3) = (\theta j(O^1D) + k_{OH+O3}[OH] + k_{HO2+O3}[HO_2] + \\ \sum(k^j_{alkene+o3}\left[ alkene^j \right]))[O_3] + k_{OH+NO2}[OH][NO_2] \qquad \text{(E3)}$$

$$net\ P(O_3) = P(O_3) - D(O_3) \qquad \text{(E4)}$$



## 3. Results and discussion

### 3.1 The correlation between PM$_{2.5}$ and AOD

In order to evaluate the extinction capacity of near-surface PM$_{2.5}$, we investigated the relationship between PM$_{2.5}$ and AOD (at 380nm). The factors that affect this relationship include aerosol type, aerosol size distribution, aerosol distribution in the vertical direction, relative humidity (RH) and planetary boundary layer height (PBLH) (van Donkelaar et al., 2010). Figure 1 shows the correlation between AOD and PM$_{2.5}$ in four different seasons. The determination coefficient ($r^2$) is 0.53, 0.58, 0.62 and 0.59 for spring (March, April and May), summer (June, July and August), autumn (September, October and November) and winter (December, January and February), respectively. Meanwhile, the correlation exhibits significant seasonal differences, having relatively smaller slope (23.56) in summer and relatively larger slope (73.76) in winter (Table 2). This implies that the aerosols in summer have stronger extinction capacity in summer than in winter. One reason for the seasonal differences is the variation in RH among different seasons. There is higher RH in summer (57.2% on average) than in winter (30.4% on average), leading to stronger hygroscopic growth of aerosol particles, and thus resulting in higher scattering ability of aerosol particles. According to another study in urban Beijing, the higher the RH, the smaller the slope, and the higher the PBLH, the smaller the slope. In addition, the slope was smaller for scattering-dominant aerosols than for absorbing-dominant





aerosols, and smaller for coarse mode aerosols than for fine mode aerosols (Zheng, C.
W et al., 2017). The slopes of the correlation between AOD (at 550nm) and $PM_{2.5}$ in
this study in summer and winter are equal to 42.2$\mu$g m$^{-3}$ and 119.2$\mu$g m$^{-3}$, respectively,
close to that from Ma et al. (2016) (54.9$\mu$g m$^{-3}$ and 110.5$\mu$g m$^{-3}$) and Xin et al. (2016)
(55.2 $\mu$g m$^{-3}$ and 93.4$\mu$g m$^{-3}$), but smaller significantly than that from Zheng et al.
(2017) (65~74$\mu$g m$^{-3}$ and 143~158$\mu$g m$^{-3}$). The differences mainly depend on the
aerosol composition and size distribution at different observational sites in Beijing.
Compared with other cities in North China (Tianjin, Shijiazhuang and Baoding) (Ma
et al., 2016), the slope in Beijing for winter is significantly higher, indicating that the
extinction capacity of aerosols in Beijing is weaker in winter.


**3.2 Seasonal and diurnal variability of AOD and photolysis frequencies**

The diurnal cycles of AOD is shown in Figure 2. AOD displays obvious diurnal

variation, with relatively high level at noon and low level at dawn and evening. The
diurnal variation of $PM_{2.5}$ is opposite to AOD. The opposite diurnal variation of AOD
and $PM_{2.5}$ is mainly due to higher development of planetary boundary layer at noon,
resulting in more complete mixture of particulate matter in the vertical direction. In
addition, AOD has obvious seasonal differences, with the highest AOD in summer
and the lowest AOD in winter. Conversely, $PM_{2.5}$ in winter (66.9$\mu$g m$^{-3}$) is
significantly higher than in summer (45.5$\mu$g m$^{-3}$). In spite of lower $PM_{2.5}$ in summer,
AOD in summer is higher due to stronger extinction capacity of $PM_{2.5}$ as discussed in





3.1. Figure 3 shows the diurnal variation of the photolysis frequencies under cloudless
conditions for each season. $j(O^1D)$ and $j(NO_2)$ are both highest in summer, followed
by spring and autumn, and lowest in winter. This seasonal difference is mainly
determined by the difference in SZA for the four seasons.

The observed photolysis frequencies at this site are lower than that observed in

the eastern Mediterranean (Crete, Greece, 35°20′N,25°40′E) (Gerasopoulos et al.,
2012) by $7.8\times10^{-6}$ s$^{-1}$ and $4.9\times10^{-6}$ s$^{-1}$ for $j(O^1D)$, and $1.9\times10^{-3}$ s$^{-1}$ and $3.3\times10^{-3}$ s$^{-1}$ for
$j(NO_2)$, in summer and winter respectively. The corresponding lower photolysis
frequencies of Beijing than the eastern Mediterranean due to SZA difference is $1.7\times$
$10^{-6}$ s$^{-1}$ and $3.0\times10^{-6}$ s$^{-1}$ for $j(O^1D)$, and $8.0\times10^{-5}$ s$^{-1}$ and $6.6\times10^{-4}$ s$^{-1}$ for $j(NO_2)$
according to TUV model, which are significantly lower than observed decreased
magnitudes. Taking into account the similar levels of ozone column concentration in
the two sites, the large gap of photolysis frequencies in the two sites is mainly
attributed to the higher AOD in Beijing (0.76±0.75) than in the eastern Mediterranean

(0.27±0.13).

It can be seen from Figure 3 that the difference between winter and summer for

$j(O^1D)$ is significantly larger than that for $j(NO_2)$, where the summer midday averages
of $j(O^1D)$ and $j(NO_2)$ are 5 times and 2 times those of winter, respectively. There are
two reasons for this phenomenon. One, compared with $j(NO_2)$, $j(O^1D)$ is more
sensitive to the change in SZA and the same change in SZA results in a larger change
in $j(O^1D)$ than $j(NO_2)$. Two, the main influential factors of $j(NO_2)$ under cloudless
conditions are SZA and AOD, and the influence of ozone column concentration and



temperature on $j(NO_2)$ is negligible. However, $j(O^1D)$ is affected significantly by the
ozone column concentration and temperature, in addition to SZA and AOD. The
higher ozone column concentration and lower temperature in winter than in summer
lead to the difference in $j(O^1D)$ further increasing.


**3.3 The correlation between photolysis frequencies and AOD**

**3.3.1 The correlation between $j(O^1D)$ and AOD**

In order to rule out the effect of SZA on photolysis frequencies, we chose SZA
equal to 30° and 60° (± 1°) for analysis. Figure 4 presents the dependence of $j(O^1D)$
on AOD at different levels of ozone column concentration at SZA of 30° and 60° (±
1°). The ozone column concentration has a classification width of 30 DU. $j(O^1D)$
exhibits a clear dependence on AOD, with a nonlinear negative correlation. As AOD
increased, the slope of $j(O^1D)$-AOD gradually decreases, indicating that the ability of
aerosols to reduce $j(O^1D)$ gradually decreases with AOD. This result differs from that
found in Mediterranean, where $j(O^1D)$ was linearly negatively correlated with AOD
(Casasanta et al., 2011; Gerasopoulos., 2012). A larger variation range of AOD in
Beijing (0-3) compared with Mediterranean (0-0.6) is one reason for the difference.
For further analysis, the observed relation between $j(O^1D)$ and AOD was
compared with TUV-simulated results. Panels a and b of Figure 5 present the
comparison between observed and TUV-simulated $j(O^1D)$ against AOD at a SZA of





30° and 60° respectively and ozone column concentration of 330-360 DU. At low
AOD level ( < 0.8), the observed slope of j($O^1D$) vs AOD is significantly larger than
the simulated slope at SSA of 0.95, and slightly larger than the simulated slope at SSA
of 0.85. With AOD increasing, the observed slope decreases rapidly to the level
smaller than the simulated slopes. The rapid change of the slope with AOD can be
related to the variation of SSA at different AOD level. Figure 6 presents the
relationship between SSA and AOD based on observed data in August 2012. The
result suggests a significant correlation between SSA and AOD. With the increase in
AOD, SSA is elevated; meanwhile, the slope of SSA vs AOD is gradually reduced.
SSA characterizes the ratio of the scattering extinction coefficient to the total
extinction coefficient (scattering extinction coefficient plus absorptive extinction
coefficient) of aerosols. The smaller the SSA, the higher the absorptive component
and lower the scattering component of the aerosol, and the stronger the ability of the
aerosol to reduce the actinic flux (Dickerson et al., 1997). Figure 6 indicates that
aerosols in Beijing under low AOD conditions had a higher proportion of absorptive
aerosol components than under high AOD conditions, and, as a result, had a stronger
ability to reduce the photolysis frequencies, which contributed to the rapidly reduced
slope of j($O^1D$) vs AOD with AOD. However, due to absence of more SSA data of the
period 2012-2015, we can't give more sufficient evidence for the dependence of SSA
on AOD. For another perspective, Owing to the biomass burning and soot emission
generated from heating, the fine mode heavily-absorbing aerosol percentage is higher
in winter than in summer (Zheng et al., 2017; Liu et al., 2016; Zhang et al., 2013), and





thus aerosols in winter have stronger ability to reduce the photolysis frequencies.
High AOD levels often appeared in summer and low AOD levels occurred mostly in
winter (Figure 2), another fact that may also explains the rapidly reduced slope of
$j(O^1D)$ vs AOD with AOD.

Comparing panels a and b of Figure 4, we see that at AOD smaller than 1, the

slope of $j(O^1D)$ vs AOD exhibits a significant dependence on SZA and the slope at 30°
of SZA is about 1.5-2.0 times larger than that at 60° of SZA. This result is similar to
that of the observations made in the central Mediterranean (Casasanta et al., 2011).
For the purpose of comparison with the study in the Mediterranean, the slope of $j(O^1D)$
vs AOD was calculated at AOD smaller than 0.7.

Table 3 presents slope, intercept and the determination coefficient ($r^2$) of linear

fits of correlation between $j(O^1D)$ and AOD for each ozone column class at AOD
smaller than 0.7. At SZA of 60° and $O_3$ column concentration of 300-330 DU, the
respective slope of the linear regression indicates a reduction of $j(O^1D)$ by $4.21 \cdot 10^{-6}$
$s^{-1}$ per AOD unit. Gerasopoulos et al. (2012) reported that the observed slope in the
eastern Mediterranean was equal to $2.44 \cdot 10^{-6}$ $s^{-1}$ at $O_3$ column of 300-320 DU.
Casasanta et al. (2011) reported that the observed slope in the central Mediterranean
varied from $2.66 \cdot 10^{-6}$ $s^{-1}$ to $3.87 \cdot 10^{-6}$ at $O_3$ column of 300-330 DU. All of these results
are smaller than the value of the present study, indicating that aerosols in urban
Beijing had a stronger extinction capacity on $j(O^1D)$ than those in the Mediterranean
that was influenced by both natural absorptive aerosols and anthropogenic aerosols.
Previous study indicated that SSA in Beijing ranged from 0.80 to 0.86 (Garland et al.,





2009;Han et al., 2015b; Han et al., 2017; Tian et al., 2015). The relatively low SSA in
Beijing could be an important reason for the stronger extinction capacity.


**3.3.2 The correlation between j(NO₂) and AOD**

Unlike $j(O^1D)$, $j(NO_2)$ is negligibly affected by ozone column concentration and

depends mainly on AOD and SZA under cloudless conditions. Figure 7 presents the
dependence of $j(NO_2)$ on AOD at different SZA levels under cloudless conditions.
The cosine of SZA (cos (SZA)) is categorized according to a width of 0.2. In the same
category of cos (SZA), $j(NO_2)$ displays a strong dependence on AOD. When cos(SZA)
is at its maximum level (0.8–1), the correlation between $j(NO_2)$ and AOD is close to
linear. When cos (SZA) decreases, the correlation tends to be nonlinear. Similar to
$j(O^1D)$, the observed slopes of $j(NO_2)$ vs AOD are also larger than TUV-simulated
slope at SSA of 0.95 and 0.85 when AOD is smaller than 0.8, and decreased rapidly
with increasing AOD (panels c and d of Figure 5). The reason for this result is the
same with that for $j(O^1D)$ as explained above.

Table 4 presents the slope, intercept and the determination coefficient ($r^2$) of

linear fits of correlation between $j(NO_2)$ and AOD for each ozone column class at
AOD smaller than 0.7. The slope of $j(NO_2)$ vs AOD also displays a significant
dependence on SZA. The slope increases as SZA increases from 0 to 0.5 and then
decreases as SZA increased from 0.5 to 1. At SZA of 60°±1 (cos (SZA)=0.5±0.015),

the respective slope of the linear regression indicates a reduction of j($NO_2$) by 3.2▪$10^{-3}$ $s^{-1}$ per AOD unit. This result is larger than the value for non-dust aerosols (2.2▪$10^{-3}$ $s^{-1}$) and close to the value for dust aerosols (3.1▪$10^{-3}$ $s^{-1}$) in the eastern Mediterranean reported by Gerasopoulos et al. (2012).

### 3.4 The parameterization relationship between photolysis frequencies, AOD, and SZA

As analyzed above, the photolysis frequencies (j($O^1D$) and j($NO_2$)) strongly depended on AOD and cos(SZA) and could be fit into expression E5 using a quadratic polynomial form. Table 5 presents the fitting parametric equations and the corresponding coefficients of determination ($R^2$) at different $O_3$ column ranges. The coefficients of determination of the fitting equations are greater than 0.95 for j($NO_2$) and j($O^1D$), indicating that both of the photolysis frequencies strongly depended on AOD and cos(SZA) at a certain $O_3$ column, and the effect of other factors such as SSA and AE are integrated into the constant term in the parametric equation. Since the ozone column concentration has greater influence on j($O^1D$) than on j($NO_2$), the parameters of fitting equations for j($NO_2$) are similar, but the parameters of fitting equations for j($O^1D$) have a large fluctuation at different $O_3$ column ranges (especially $a_1$ and $a_2$). The parametric equations can be used to quantitatively evaluate the effect of AOD on photolysis frequencies in Beijing. According to the parametric equations,





aerosols lead to a decrease in j($NO_2$) by 24.2% and 30.4% and a decrease in j($O^1D$) by
27.3% and 32.6% in summer and winter, respectively, compared to an aerosol-free
atmosphere. The decreasing ratio of the photolysis frequencies in winter is higher than
in summer due to the higher SZA and lower SSA in winter.

The effect of aerosols on photolysis frequencies in Beijing is compared with

other studies. Real and Sartelet (2011) reported a reduction in j($NO_2$) and j($O^1D$) of
13%-14% due to aerosols by using the radiative transfer code Fast-J during summer
2001 over European regions. Flynn et al (2010) reported that aerosols reduced j($NO_2$)
by 3% in Huston during 2006 by using TUV model. Gerasopoulos et al (2012)
reported that aerosols reduced j($NO_2$) and j($O^1D$) by 5%-15% with 5-yr mean AOD at
380nm equal to 0.27. All of these results are lower than the reduction ratio of this
study mainly due to higher aerosol level in Beijing (4-yr mean AOD equal to 0.76±
0.75). Hodzic et al. (2007) simulated a 15–30% j($NO_2$) photolysis reduction during
the 2003 European summer heatwave in the case of absorbing biomass burning
aerosols with AOD at 550 nm equal to 0.7-0.8 and SSA at 532 nm equal to 0.83-0.87.
The result of this study is comparable with the reduction ratio of this study possibly
due to the equivalent levels of AOD and SSA. In addition, Péré et al (2015) simulated
a higher reduction (20–50%) in j($NO_2$) and j($O^1D$) along the transport of the aerosol
plume during the 2010 Russian summer wildfires episode. The higher reduction is due
to the higher level of AOD (peak value of AOD at 400nm reached 2-4), even though
SSA is very high (0.97).
$j = a_1 + a_2 AOD + a_3 cos(SZA) + a_4 (AOD)^2 + a_5 AOD cos(SZA) + a_6 (cos(SZA))^2 \ldots E5$




The above established parametric relationship of $PM_{2.5}$-AOD and

$j(NO_2)$-AOD-SZA gives us a chance to estimate the effect of $PM_{2.5}$ on photolysis

frequencies due to aerosol light extinction.

**3.5 The influence of AOD on ozone production**

In order to explain the effect of aerosol light extinction on ozone production, we

used the data from the field observation campaign undertaken in August 2012. Ozone

production depends on its precursors (NOx and VOCs), meteorological factors, and

solar radiation. Solar radiation is the driving force for tropospheric photochemical

reactions, in which $j(O^1D)$ and $j(NO_2)$ are both important for ozone production. On

the one hand, the increase in $j(NO_2)$ promotes the photolysis of $NO_2$, thereby

accelerating the formation of ozone. On the other hand, the increase in $j(O^1D)$

accelerates the photolysis of ozone. In addition, the increase in the photolysis

frequencies will accelerate the photolysis of OVOC (especially formaldehyde and

acetaldehyde), HONO, and $H_2O_2$, resulting in increases in OH and $HO_2$, which will

promote the reaction between OH and VOCs and thus produce more $RO_2$. As a result,

more ozone is produced by increasing the reaction rate between $RO_2$ (or $HO_2$) and NO.

However, the increase in OH and $HO_2$ also consumes ozone and $NO_2$, which

contributes to the increase in $D(O_3)$. In brief, the overall effect of changes in



photolysis frequencies on sources and sinks of ozone determines the change in the net
ozone production rate.

Ozone production ($HO_2$ + NO, $RO_2$ + NO), ozone loss ($O^1D$ + $H_2O$, $HO_2$ + $O_3$,

$O_3$ + OH, $NO_2$ + OH, and $O_3$ + alkenes), and net ozone production rate during August
2012 were calculated by using the box model. We used the observed photolysis
frequencies (i.e. j_obs) and the calculated photolysis frequencies by parametric
equation under the condition of AOD equal to 0 (i.e. j_AOD=0), were used to
constrain the box model. The difference of simulated results in the two scenarios can
be attributed to the effect of aerosol light extinction. As a result, the presence of
aerosols causes a decrease in both ozone production rate and loss rate, as is shown in
Figure 8. Since the decreasing amplitude of the ozone production rate is far larger
than that of the ozone loss rate, the net production rate of ozone is reduced by 25%.
This reduction is comparable with the results of the study in Mexico City, where
aerosols caused a 20% reduction in the ozone concentrations (Castro et al., 2001).
Studies in Houston and Crete have shown that aerosols cause ozone production rates
to decrease by about 4% and 12%, respectively, which are lower than that found in
this study (Flynn et al., 2010; Gerasopoulos et al., 2012).

The ratio of the observed photolysis frequencies to the photolysis frequencies at

AOD equal to 0 is defined as JIF (Flynn et al., 2010). A JIF of less than 1 indicates
that the aerosols cause a decrease in the photolysis frequencies. Figure 9 shows the
relation between $P(O_3)_{j\_obs}/P(O_3)_{j\_AOD=0}$ (or $D(O_3)_{j\_obs}/D(O_3)_{j\_AOD=0}$) and JIF. The
majority of JIF values were less than 1, with an average of 0.72, indicating that



aerosols greatly attenuated photolysis frequencies due to high levels of AOD (average
of 1.07) and low levels of SSA (average of 0.84) during the observation period.
$P(O_3)_{j\_obs}/P(O_3)_{j\_AOD=0}$ and $D(O_3)_{j\_obs}/D(O_3)_{j\_AOD=0}$ are both linearly positively
correlated with JIF and the scatters are mostly above the 1:1 line. As can be seen from
the figure 9, a 30% reduction in photolysis frequencies (JIF = 0.7) due to the presence
of aerosols results in a decrease in ozone production rate and loss rate by about 26%
and 15%, respectively. The decreasing amplitude in the ozone production rate is
greater than the decrease in the ozone loss rate because the corresponding processes
of ozone production are all light-driven, but the corresponding processes of ozone loss
are not all light-driven because the reaction of $O_3$ with alkenes does not depend on
solar radiation. According to the simulated results, the reaction of ozone with alkenes
during this campaign accounts for 17% of total ozone loss.

The diurnal profile of the mean ozone production and loss rate is shown in

Figure 10. $P(O_3)$ peak midday in the 12:00-14:00 local hours at 31 ppb/h without
aerosol impact and 23 ppb/h with aerosol impact. The maximum $D(O_3)$ also occurrs
between 12:00 and 14:00 at 4.2 ppb/h without aerosol impact and 3.5 ppb/h with
aerosol impact. There is little difference between aerosol-impact and aerosol-free
$P(O_3)$ (or $D(O_3)$) in the hours of 6:00-11:00, but the difference in the afternoon
(12:00-18:00) is large, indicating that the reduction effect of aerosol on ozone
production mainly occurs during the afternoon.

The above analysis focuses on the effect of aerosol on the ozone production due

to aerosol light extinction. However, it does not consider the close relationship





between aerosol and ozone's gaseous precursors in the actual atmosphere. To explain
this problem, we chose two adjacent days (small SZA effect) with obviously different
AOD levels: a clean day (A day: August 21, 2012; AOD = 0.21, $PM_{2.5}$=21.6 µg m$^{-3}$)
and a day with high aerosol pollution (B day; August 26, 2012; AOD = 3.2,
$PM_{2.5}$=125.0 µg m$^{-3}$) (Table 7). The difference in AOD between the two days can be
taken to represent the maximum daytime gap of AOD for this month. The ozone
column concentrations for these two days were 302 DU and 301 DU, respectively, of
which the effect on $j(O^1D)$ is negligible. Under these conditions, the $j(O^1D)$ value at
noon time decreases from $3.23 \times 10^5$ s$^{-1}$ on A day to $1.29 \times 10^5$ s$^{-1}$ on B day (i.e., a 60%
reduction) and the $j(NO_2)$ value at noon time decreases from $8.26 \times 10^{-3}$ s$^{-1}$ on A day
to $4.19 \times 10^{-3}$s$^{-1}$ on B day (i.e., a 49.2% reduction). As shown in Table 7, B day has
higher AOD and higher concentrations of gaseous pollutants. The concentrations of
CO, $NO_2$, HCHO and the OH reactivity of VOCs in B day are much higher than in A
day, with the ratio of 3.6, 2.3, and 2.0, respectively. The simultaneous increases of
gaseous pollutants and AOD are due to the fact that gaseous pollutants ($NO_x$, $SO_2$,
and VOCs) emitted by major pollution sources in Beijing, including traffic and
industry, have undergone the processes of gas-phase oxidation and nucleation to
generate secondary particulate matter that contributes to aerosol light extinction.
Previous studies have reported that secondary particulate matter has accounted for
more than 60% of total particulate matter during severe smog pollution in Beijing
summers (Han et al., 2015a; Guo et al., 2014). In addition, several studies have shown
that secondary components in particulate matter (especially secondary organics and



ammonium sulfate) have dominated the aerosol light extinction (Han et al., 2014; Han
et al., 2017; Wang et al., 2015). Observations made in Beijing during the summer of
2006 showed that ammonium sulfate and ammonium nitrate contributed 44.6% and
22.3%, respectively, to the total extinction coefficient during a severe period of smog
(Han et al., 2014); in the summer of 2014 in Beijing, ammonium sulfate, secondary
organic aerosols, and ammonium nitrate contributed 30%, 22%, and 18%, respectively,
to the total extinction coefficient (Han et al., 2017).

As shown in Figure 11, the simulation results indicate that the net $P(O_3)$ of B day

is 36.2% higher than that of A day due to higher concentrations of ozone precursors
on B day. This result is consistent with the observed ozone concentrations, of which
the observed ozone concentration in B day is 2.2 times higher than that of A day. If we
adjust the photolysis frequencies level of B day to the level of A day, the net $P(O_3)$
increases by 70.0%, which indicates that the high level of particulate matter in B day
greatly inhibits ozone production. This result means that the system is under negative
feedback, thus keeping $O_3$ at a relatively stable level. Table 8 summarizes the average
levels of gaseous pollutants and photolysis frequencies for AOD less than 1 and
greater than 1, as measured during August 2012. It shows that, the concentrations of
ozone's precursors are higher and the photolysis frequencies are lower at high AOD
levels (AOD > 1) than those at low AOD level (AOD < 1). This result means that the
negative feedback mechanism is prevalent throughout the whole campaign period.
Therefore, the prevention and control measures of air pollution in Beijing need to
incorporate this coupling mechanism between particulate matter and ozone to achieve



effective control of these two main pollutants.

**4. Conclusion**

Photolysis reactions are important driving forces for tropospheric photochemical

oxidation processes and ozone production. In this study, we explored in detail the
effects of aerosols on photolysis frequencies and ozone production in Beijing, based
on a long observation period of 4 years. We have found that:

(1) There is a strong correlation between $PM_{2.5}$ and AOD, and the slope in

summer is smaller significantly than in winter, which indicates that aerosols

in summer have a more efficient extinction capacity than in winter.

(2) As AOD increased, the extinction effect of aerosol on photolysis frequencies

was decreased; this result was related to a higher proportion of scattering

aerosols under high AOD conditions than under low AOD conditions. The

slope of the correlation between photolysis frequencies and AOD indicates

that the aerosols in urban Beijing have a stronger extinction on actinic flux

than absorptive dust aerosols in the Mediterranean.

(3) The influence of AOD on photolysis frequencies was evaluated quantitatively

by establishing parametric equations. According to the parametric equation,

aerosols lead to a decrease in $j(NO_2)$ by 24.2% and 30.4% for summer and

winter, respectively, and the corresponding decrease in $j(O^1D)$ by 27.3% and

32.6% respectively, compared to an aerosol-free atmosphere.



(4) In order to evaluate the effects of aerosols on ozone production rate, we

carried out an observation campaign in August 2014. The results show that

aerosols reduced the net ozone production rate by 25% by reducing the

photolysis frequencies. High concentrations of ozone gaseous precursors

were often accompanied by high concentrations of particulate matter, which,

to a large extent, inhibited excessive levels of ozone generation and reflected

the negative feedback effect of the atmospheric system. Therefore, the

influence of aerosol on photolysis frequencies and thus on the rate of

oxidation of VOCs and NOx to ozone and secondary aerosol is important for

determining the atmospheric effects of controlling the precursor emissions of

these two important air pollutants (aerosols and ozone).


**Author contribution**

| Author | Contribution |
| --- | --- |
| Wenjie Wang | acquisition of data; analysis and interpretation of data; drafting the article and revising it critically |
| Min Shao | substantial contributions to conception and design; revising the article critically |
| Min Hu | collection of data |
| Limin Zeng | collection of data |
| Yusheng Wu | collection of data |








**ACKNOWLEDGEMENTS**

This work was supported by the Major Program of the National Natural Science Foundation of China [Grant number 91644222]. We thank Hongbin Chen and Philippe Goloub for data management of AOD and other aerosol optical properties on AERONET.


















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









Table 1. Instruments deployed in the field campaign undertaken in August 2012 and
used for data analysis.

| Parameters | Measurement technique | Time resolution | Detection limit | Accuracy |
|---|---|---|---|---|
| $j(O^1D)$ and $j(NO_2)$ | Spectroradiometer | 10 s | / | ± 10% |
| $O_3$ | UV photometry | 60 s | 0.5 ppbv | ± 5% |
| NO | Chemiluminescence | 60 s | 60 pptv | ± 20% |
| $NO_2$ | Chemiluminescence | 60 s | 300 pptv | ± 20% |
| CO | IR photometry | 60 s | 4 ppb | ± 5% |
| $SO_2$ | Pulsed UV fluorescence | 60 s | 0.1 ppbv | ± 5% |
| HCHO | Hantzsch fluorimetry | 60 s | 25 pptv | ± 5% |
| VOCs | GC-FID/MS | 1 h | 20-300 pptv | ± 15~20% |



















Table 2. O$_3$ column concentration, temperature relative humidity and respective
standard deviation for different seasons.

| | O$_3$ column (Du) | Temperature (℃) | Relative humidity (%) |
|---|---|---|---|
| Spring | 354.9±37.3 | 15.6±7.8 | 33.2±18.1 |
| Summer | 310.2±23.8 | 27.5±4.2 | 57.2±17.7 |
| Autumn | 303.8±22.8 | 15.5±7.4 | 46.4±20.6 |
| Winter | 347.2±28.2 | 0.53±4.24 | 30.4±17.6 |























Table 3. Slope, intercept and the square of correlation coefficient ($r^2$) of linear fits of
correlation between $j(O^1D)$ and AOD for each ozone column class at AOD smaller
than 0.7.

| | SZA=30° | | | SZA=60° | | |
|---|---|---|---|---|---|---|
| $O_3$ column (DU) | Slope ($10^{-6}s^{-1}$) | Intercept ($10^{-6}s^{-1}$) | $r^2$ | Slope ($10^{-6}s^{-1}$) | Intercept ($10^{-6}s^{-1}$) | $r^2$ |
| 300-330 | -6.24±1.52 | 25.7±0.8 | 0.34 | -4.21±0.43 | 7.67±0.33 | 0.41 |
| 330-360 | -6.50±1.43 | 23.2±0.6 | 0.40 | -5.01±0.34 | 7.15±0.21 | 0.52 |
| 360-390 | -9.45±1.64 | 20.9±0.9 | 0.52 | -6.93±0.62 | 7.59±0.34 | 0.66 |





















Table 4. Slope, intercept and the square of correlation coefficient ($r^2$) of linear fits of
correlation between $j(NO_2)$ and AOD for each ozone column class at AOD smaller
than 0.7.

| cos(SZA) | Slope ($10^{-3}$ s$^{-1}$) | Intercept ($10^{-3}$ s$^{-1}$) | $r^2$ |
|---|---|---|---|
| 0-0.2 | -1.28±0.07 | 1.54±0.04 | 0.52 |
| 0.2-0.4 | -2.44±0.10 | 3.40±0.04 | 0.41 |
| 0.4-0.6 | -3.20±0.09 | 5.49±0.04 | 0.49 |
| 0.6-0.8 | -2.08±0.09 | 7.20±0.05 | 0.38 |
| 0.8-1.0 | -1.77±0.12 | 8.12±0.05 | 0.26 |





















Table 5. The fitting parameters $a_1$-$a_6$ and determination coefficients of E5 for $j(NO_2)$.

| $O_3$ column (DU) | $a_1$ | $a_2$ | $a_3$ | $a_4$ | $a_5$ | $a_6$ | $r^2$ |
|---|---|---|---|---|---|---|---|
| | | | $\times 10^{-3}$ | | | | |
| 270-300 | $-0.20 \pm 0.09$ | $-2.1 \pm 0.1$ | $13.1 \pm 0.4$ | $0.27 \pm 0.02$ | $0.19 \pm 0.09$ | $-3.5 \pm 0.3$ | 0.96 |
| 300-330 | $-0.48 \pm 0.07$ | $-1.9 \pm 0.1$ | $13.3 \pm 0.3$ | $0.19 \pm 0.01$ | $0.34 \pm 0.08$ | $-3.9 \pm 0.3$ | 0.96 |
| 330-360 | $-0.22 \pm 0.08$ | $-2.2 \pm 0.1$ | $11.8 \pm 0.3$ | $0.42 \pm 0.03$ | $0.23 \pm 0.03$ | $-2.6 \pm 0.2$ | 0.96 |
| 360-400 | $-0.21 \pm 0.10$ | $-2.0 \pm 0.1$ | $12.6 \pm 0.2$ | $0.18 \pm 0.02$ | $0.39 \pm 0.03$ | $-4.0 \pm 0.3$ | 0.95 |


Table 6. The fitting parameters $a_1$-$a_6$ and determination coefficients of E5 for $j(O^1D)$.

| $O_3$ column (Du) | $a_1$ | $a_2$ | $a_3$ | $a_4$ | $a_5$ | $a_6$ | $r^2$ |
|---|---|---|---|---|---|---|---|
| | | | $\times 10^{-6}$ | | | | |
| 270-300 | $0.88 \pm 0.30$ | $-0.10 \pm 0.21$ | $-5.1 \pm 0.5$ | $0.93 \pm 0.06$ | $-8.6 \pm 0.4$ | $43.9 \pm 0.8$ | 0.96 |
| 300-330 | $0.58 \pm 0.07$ | $0.13 \pm 0.17$ | $-3.8 \pm 0.8$ | $0.68 \pm 0.04$ | $-7.1 \pm 0.2$ | $37.1 \pm 0.8$ | 0.96 |
| 330-360 | $2.2 \pm 0.20$ | $-0.65 \pm 0.19$ | $-9.8 \pm 0.9$ | $1.01 \pm 0.07$ | $-6.3 \pm 0.3$ | $38.1 \pm 0.6$ | 0.97 |
| 360-400 | $2.0 \pm 0.10$ | $-0.72 \pm 0.40$ | $-7.0 \pm 0.5$ | $0.76 \pm 0.08$ | $-6.2 \pm 0.7$ | $33.0 \pm 0.8$ | 0.95 |












Table 7. Mean and standard deviation of observed data during daytime (6:00–18:00)
for A day and B day.

| Observed data | A day: August 21, 2012 | B day: August 26, 2012 |
|---|---|---|
| AOD | $0.21 \pm 0.05$ | $3.2 \pm 0.4$ |
| PM$_{2.5}$ ($\mu g\ m^{-3}$) | $21.6 \pm 9.0$ | $125.0 \pm 15.7$ |
| O$_3$ column (Du) | $302 \pm 3.0$ | $301 \pm 3.0$ |
| Temperature(°C) | $27.6 \pm 3.3$ | $27.6 \pm 3.2$ |
| Relative humidity (%) | $47.6 \pm 10.1$ | $54.5 \pm 11.8$ |
| j(O$^1$D)(s$^{-1}$) | $1.57 \times 0^{-5} \pm 1.24 \times 10^{-5}$ | $6.87 \times 10^{-6} \pm 5.2 \times 10^{-6}$ |
| j(NO$_2$)(s$^{-1}$) | $5.37 \times 10^{-3} \pm 2.88 \times 10^{-3}$ | $2.87 \times 10^{-3} \pm 1.65 \times 10^{-3}$ |
| O$_3$ (ppb) | $39.7 \pm 16.56$ | $86.8 \pm 52.82$ |
| NO$_2$ (ppb) | $10.7 \pm 5.0$ | $24.9 \pm 9.6$ |
| CO (ppm) | $0.24 \pm 0.05$ | $0.85 \pm 0.14$ |
| VOC reactivity (s$^{-1}$) | $3.0 \pm 0.7$ | $6.4 \pm 1.7$ |
| HCHO (ppb) | $2.7 \pm 1.1$ | $7.4 \pm 1.9$ |









Table 8. Monthly mean and standard deviation of observed data during daytime
(6:00–18:00) under the condition of AOD less than 1 and larger than 1 in August 2012

| Observed data | AOD<1 | AOD>1 |
|---|---|---|
| AOD | $0.43 \pm 0.24$ | $2.0 \pm 0.8$ |
| $PM_{2.5}$ ($\mu g\ m^{-3}$) | $26.4 \pm 12.4$ | $76.9 \pm 47.1$ |
| $O_3$ column (Du) | $303 \pm 4.0$ | $302 \pm 5.0$ |
| Temperature(°C) | $29.6 \pm 4.3$ | $29.2 \pm 4.1$ |
| Relative humidity (%) | $42.1 \pm 15.8$ | $57.0 \pm 12.8$ |
| $j(O^1D)(s^{-1})$ | $1.62 \times 10^{-5} \pm 1.05 \times 10^{-5}$ | $1.03 \times 10^{-5} \pm 0.67 \times 10^{-5}$ |
| $j(NO_2)(s^{-1})$ | $5.64 \times 10^{-3} \pm 2.42 \times 10^{-3}$ | $3.80 \times 10^{-3} \pm 1.66 \times 10^{-3}$ |
| $O_3$ (ppb) | $52.4 \pm 33.8$ | $67.9 \pm 45.7$ |
| $NO_2$ (ppb) | $16.4 \pm 7.8$ | $24.4 \pm 8.9$ |
| CO (ppm) | $0.47 \pm 0.20$ | $0.95 \pm 0.47$ |
| VOC reactivity ($s^{-1}$) | $4.3 \pm 1.7$ | $6.2 \pm 2.2$ |
| HCHO (ppb) | $4.0 \pm 1.4$ | $6.5 \pm 1.9$ |












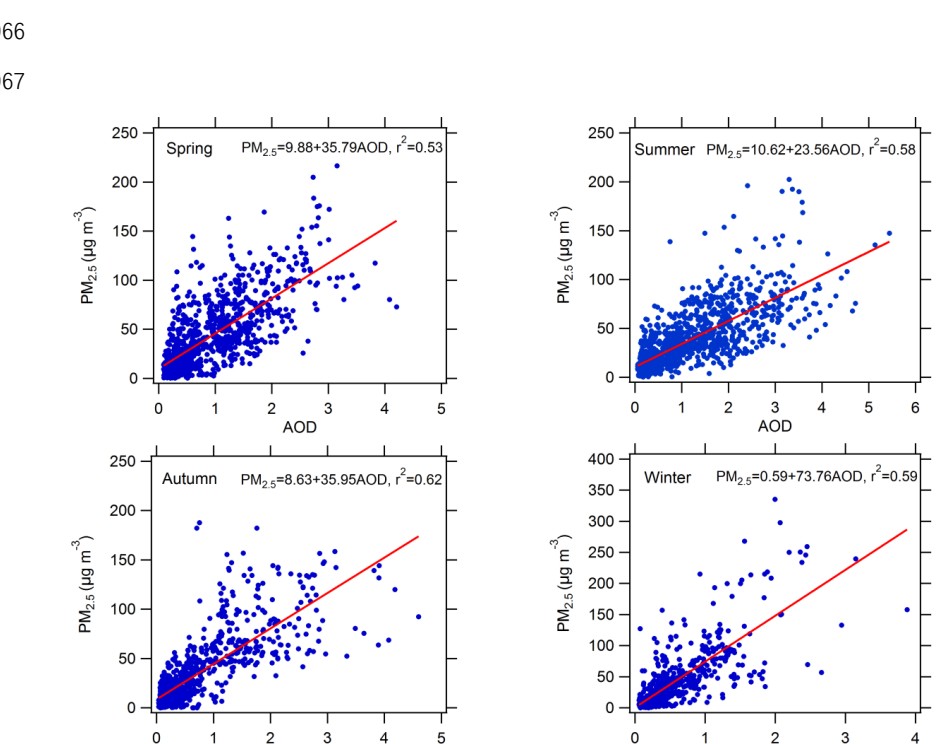


Figure 1. Scatter plots between AOD at 380nm and $PM_{2.5}$ in four different seasons.

The slope, intercept and determination coefficient ($r^2$) were calculated.


















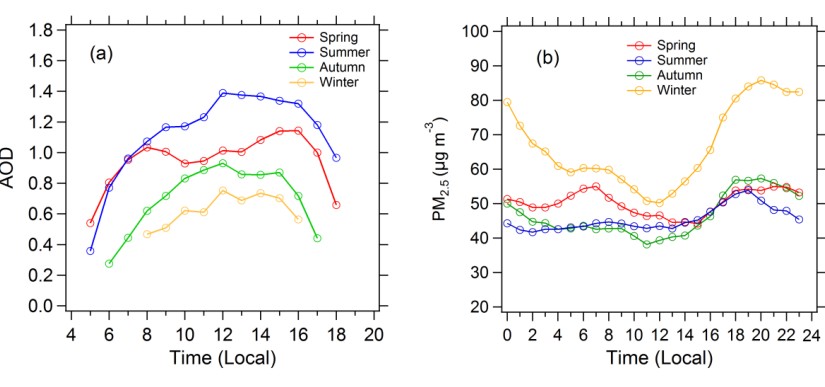


Figure 2. Diurnal cycles of (a) AOD and (b) $PM_{2.5}$ in the four seasons.


























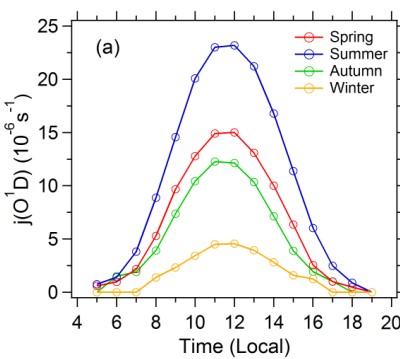 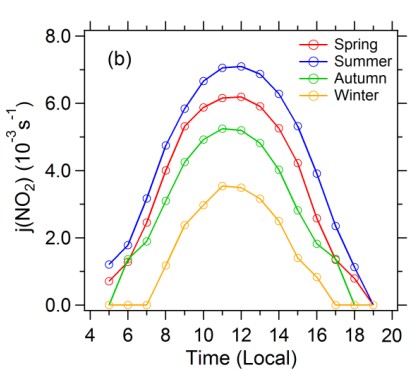


Figure 3. Diurnal cycles of (a) $j(O^1D)$ and (b) $j(NO_2)$ in the four seasons under
cloudless conditions.























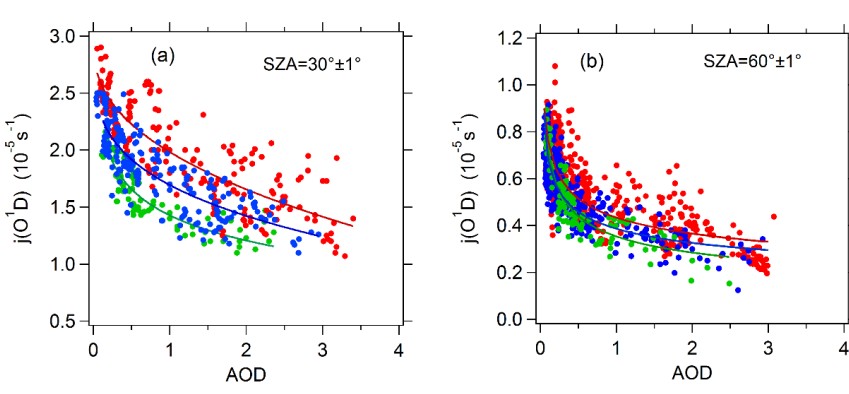


Figure 4. Dependence of j(O$^1$D) on AOD at SZA of (a) 30° and (b) 60° and at
different classes of ozone column concentration: 300-330 DU (red), 330-360 DU
(blue), and 360-390 DU (green).


















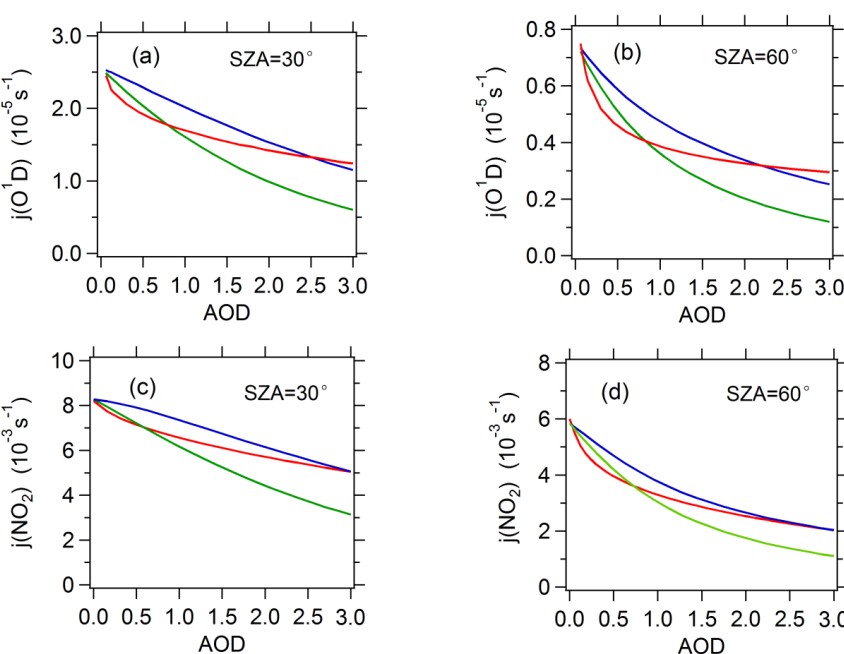


Figure 5. The relationship between observed or TUV-simulated photolysis

frequencies and AOD at SZA of 30° and 60°. The red line represents observed

average photolysis frequencies; the blue line and green line represents TUV-simulated

photolysis frequencies at SSA of 0.95 and 0.85 respectively.















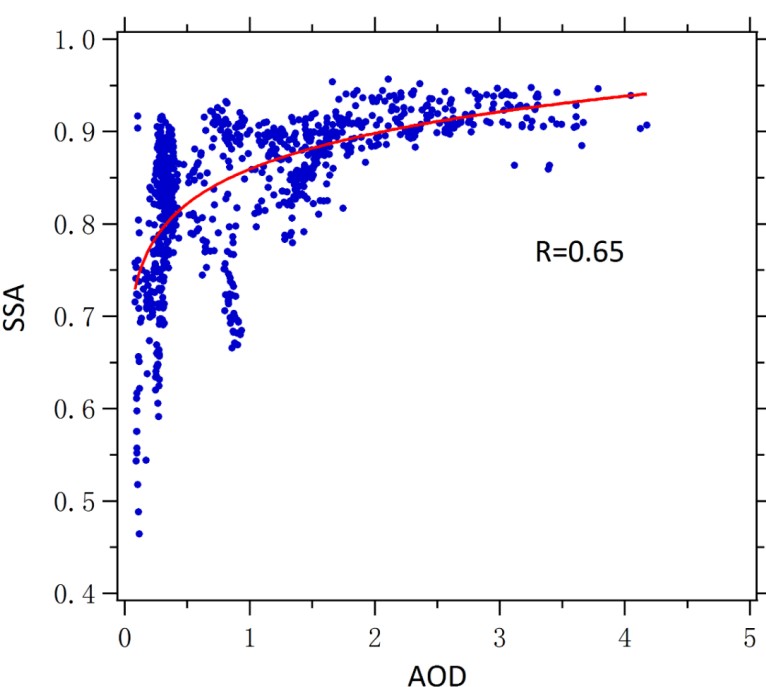


Figure 6. Correlation between SSA and AOD observed in August 2012.

















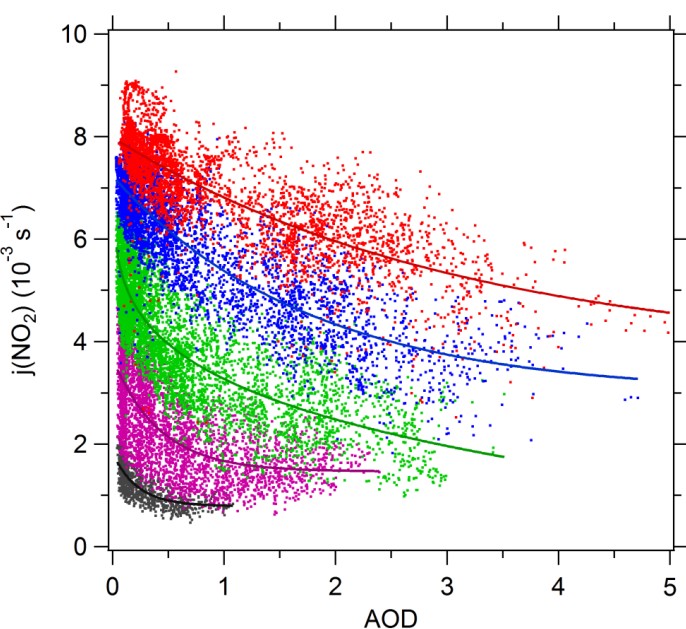


Figure 7. Dependence of j(NO$_2$) on AOD at different SZA classes. The classes of
cos(SZA) are 0–0.2 (black), 0.2–0.4 (purple), 0.4–0.6 (green), 0.6–0.8 (blue), and
0.8–1 (red).















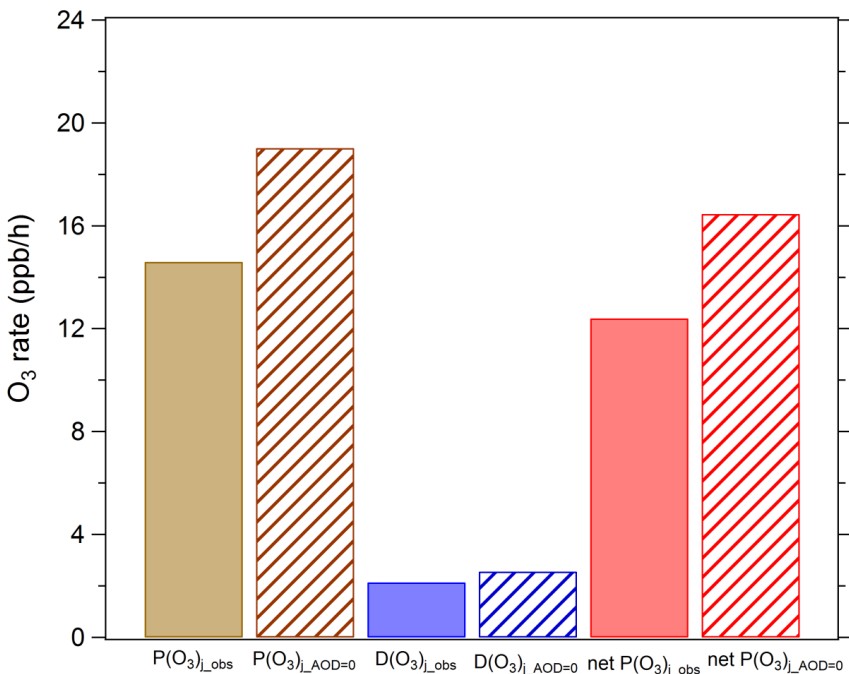


Figure 8. Average ozone production and loss terms in August 2012. $P(O_3)_{j\_obs}$,
$D(O_3)_{j\_obs}$ and net $P(O_3)_{j\_obs}$ represents ozone production rate, ozone loss rate, and net
ozone production rate under observed photolysis frequencies; $P(O_3)_{j\_AOD=0}$, $D(O_3)$
$_{j\_AOD=0}$ and net $P(O_3)_{j\_AOD=0}$ represents ozone production rate, ozone loss rate, and net
ozone production rate under calculated photolysis frequencies when AOD is equal to

1106     0.











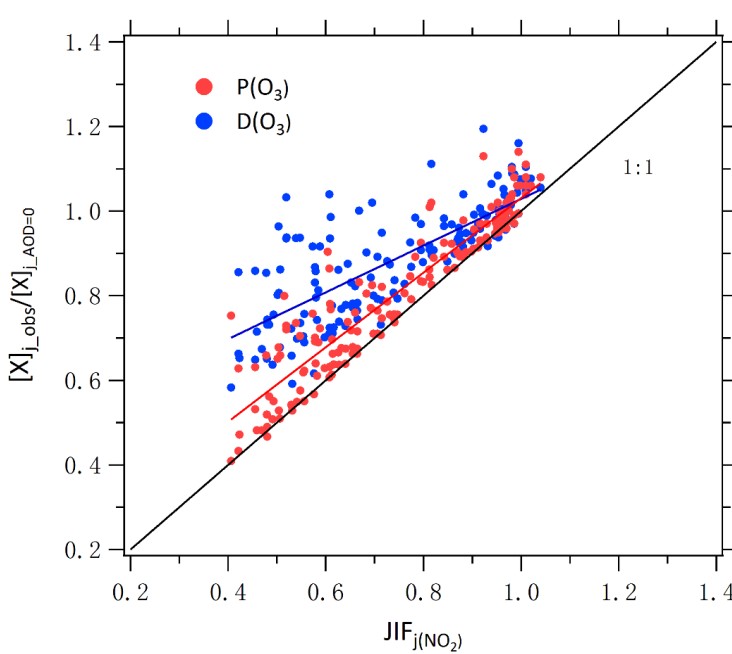


Figure 9. Correlation between $P(O_3)_{j\_obs}/P(O_3)_{j\_AOD=0}$ (or $D(O_3)_{j\_obs}/D(O_3)_{j\_AOD=0}$) and
JIF of $j(NO_2)$.

















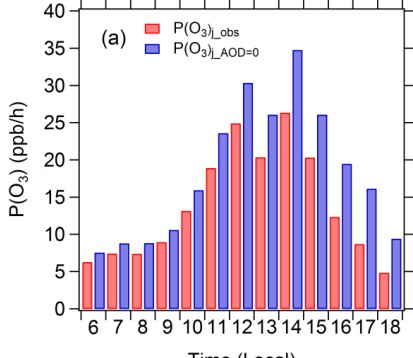
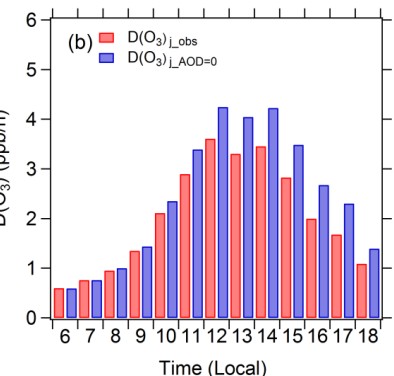


Figure 10. Diurnal profiles of mean $P(O_3)_{j\_obs}$, $P(O_3)_{j\_AOD=0}$, $D(O_3)_{j\_obs}$, and
$D(O_3)_{j\_AOD=0}$.

























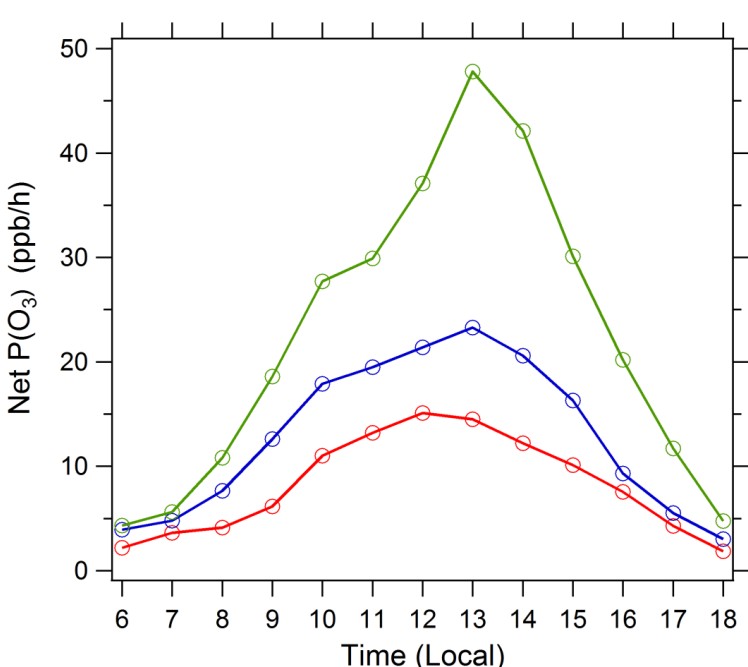


Figure 11. Diurnal profile of net $P(O_3)$ simulated by the box model. Three cases are
displayed: (1) A day (red circles); (2) B day (blue circles); and (3) the photolysis
frequencies of B day adjusted to the level of A day with other conditions unchanged
(green circles).