# Peer review of "The impact of aerosols on photolysis frequencies and ozone"

_Atmospheric Chemistry and Physics, 2019_

## Referee Comment (RC1) · Anonymous Referee #2 · 8 Apr 2019

Comments on "The impact of aerosols on photolysis frequencies and ozone production in urban Beijing during the four-year period 2012–2015"

This paper is investigating the effect of aod on J(O1D) and J(NO2) photolysis rates, considering not only the significance of photolysis rates in atmospheric chemistry but also taking into account the continuously increasing pollution in the area of Bejing. The results demonstrate the stronger extinction of actinic flux (and photolysis rates) due to aerosols in urban Bejing compared to other sites The paper is clearly written, the methodology is well understood and the results are well summarized

COMMENTS

1) In the introduction, after the definition of actinic flux (line 68) the authors could include that since the photolysis rates are proportional to the actinic flux and not all stations acquire a $2\pi$ spectroradiometer or chemical actinometers for J measurements, several methods have been developed to determine actinic flux and photolysis rates from ground based measurements of irradiance (Kylling et al 2003, Kazadzis al. 2000, 2004, Topaloglou et al. 2005, Trebs et al. 2009).

2) It is stated, in the abstract, that the reduction of J(O1D) and J(NO2) is in the order of 24.2% and 30.4% (for summer and winter respectively) while for the J(NO2) in the order of (27.3 an 32.6%) compared to an aerosol free atmosphere (aod=0?). Since the parametric equations include sza and AOD, the authors could clarifiy how exactly these percentages have been calculated i) to what sza are these percentages referring to? Also for what ozone class for J(O1D)? ii) are these maximum reductions for maximum aod observed or for a mean aod value (i.e. 0.76 )? iii) Through which parameters are summer and winter percentages calculated?

3) How do the authors comment the (low) r2 coefficient in the linear fits of of J(O1D) and J(NO2) versus aod for aod<0.7?

4) Concerning the TUV radiation model, information (apart from ssa values) about the input that was used could be included, such as solar spectrum used, aerosol profile etc. In p.2.2 it is stated that global irradiance spectra are calculated. Do you maybe mean actinic flux spectra? Since photolysis rates are proportional to actinic flux, has any comparison been done between the actinic flux measured by the spectroradiometer and that from the TUV model in order to demonstrate the level of agreement?

5) In line 419, the enhanced aerosol level in Bejing is quantified (4-year mead aod = 0.76±0.76). Some references to the studies should be included

6) In Line 254: "….according to another study in urban Bejing, ..", the reference of the study should be included.

7) Figure 6: Similar results have been obtained by Bais et al., 2005, Krotkov et al., 2004 and Kazadzis et al., 2017). Is this AOD -SSA dependence from August 2012 obvious during all seasons ? For which wavelength are SSA values given? As both parameters have a wavelength dependence and since PF ozone "effective" wavelengths are ~305-315nm, could this dependence play some role in the provided analysis of the AOD and SSA effects on PFs. ?

8) Figures 4 and 7 : Some commentation on the scatter of J's would be helpful Technical corrections

Line 249-250: Repetition of "in summer" "This implies that the aerosols in summer have stronger extinction capacity in summer than in winter"

Lines 384 &385: cos(SZA) instead of SZA

Line 423: "..The result of this study is comparable to the reduction ratio of this study possibly due to..". Probably the one "this study" refers to the previous study mentioned, Hodzie et el. 2007 and the second one to the auhors study, it would be helpful to rephrase.

Line 559: ". . .in August 2014..", refers to the field campaign in August 2012, mentioned in the paper.
* * *

---

## Referee Comment (RC2) · Anonymous Referee #1 · 8 Apr 2019

The work by Wang et al. studies the reducing impact of aerosols on photolysis frequencies j(O1D) and j(NO2) during a four-year measurement period in Beijing. Aerosol optical depths (AOD) were taken from AERONET measurements and photolysis frequencies were derived from co-located radiometric measurements of spectral actinic flux. Based on these measurements, polynomial parametrizations were derived describing empirically the influence of AOD on photolysis frequencies j(O1D) and j(NO2) in Beijing (under clear-sky conditions). The parameterizations were then used to estimate the effect of aerosol-diminished photolysis frequencies on photochemical ozone production in a chemical box model and compared to hypothetical aerosol free conditions. A 25% reduction of the monthly mean ozone production was predicted for a

selected summer month of the year 2012.

The work describes an interesting data set and comes to interesting conclusions. However, although the text excessively describes many details, some important information is missing and the data analysis is in part not satisfying. The manuscript needs major revision before publication can be recommended. The text should be widely shortened and focused. Moreover, the specific comments listed below should be addressed.

Specific comments

Abstract

Line 30: "In addition, the slopes are equal to ...." 1) The wording: "In addition, ..." is awkward. 2) Why there is a range of slopes for j(O1D) and a single value for j(NO2) is unclear at this point, and what the slopes refer to in the first place when the relationships are non-linear. 3) The slopes should be negative in any case. 4) The authors should take into account significant digits (throughout the paper). The precision of the data does not justify a statement "4.21-6.93". I would say "4.2-6.9" at the very most. 5) AOD has to be specified here, i.e. AOD (380 nm)?

Line 32: "... larger than those observed in the Mediterranean." I would say: "... than those observed in a similar, previous study in the Mediterranean."

Line 33: "...have a stronger extinction on ..." Please reword.

Line 38, 39: "... j(NO2 ) by 24.2% and 30.4% for summer and winter, ... j(O1D) by 27.3% and 32.6%..." 1) The meaning of these numbers is unclear. I assume they refer to some kind of seasonal mean of the photolysis frequencies that needs to be specified. 2) The precision implied by three digits is misleading.

Line 42: "... the monthly average net ozone production is reduced by 25%." By looking at Fig. 10, I assume the 25% refers to a monthly mean daytime net ozone production that needs to be specified.

Introduction

Lines 54, 59, 63, 65, 66: Use consistent notations for O(3P) and O(1D).

Line 60: "....the only significant chemical source...."

Lines 68-71: Remove the symbol "S" in the brackets. It may be added as an index to "sigma" and "phi" but "S" is no variable like "lambda" or "T".

Lines 78-79: "Scattering aerosols can enhance..., while absorptive aerosols reduce ...throughout the boundary layer." These statements are unclear and certainly do not apply for all conditions.

Line 91: "Therefore it is necessary to quantitatively evaluate the effect of aerosols on photolysis frequencies for the purpose of ozone prevention". I would say: "... for a better understanding of ozone formation under highly polluted conditions."

Line 98: "... are compared with the observed value to test the simulation effect." Unclear: If radiative transfer models are used there are usually no measurements available. And what is the "simulation effect"?

Line 100: "... due to complicated environmental conditions...". Clarify.

Line 123-136: Use the term "photolysis frequencies" consistently throughout the text.

Lines 134, 135: Mind significant digits, see abstract by Li et al., 2011.

Line 143ff: "Our overall goal..." It should be made clear that this study was strongly informed by a similar work by Gerasopoulos et al., 2012 which is not adequately referred to in the Introduction.

Measurements

Line 155: The exact measurement period should be given here. Was it exactly four years?

Line 163: What absorption cross sections and quantum yields were used to calculate

the photolysis frequencies?

Line 163: I assume the j(O1D) were calculated temperature dependent according to Eq. 1 (and a statement in lines 302-304). That should be clearly stated. However, is it useful, if a common parameterization as a function of AOD is later used for summer and winter j(O1D)? There may be arguments to include temperature but the influence should be mentioned and quantified in Section 3 (see below).

Line 164: "Shetter and Müller , 1999"

Line 169: The studies by Shetter and Müller, 1999, and Hofzumahaus et al., 1999 describe double-monochromator based instruments with somewhat different properties. The authors should state what type of detector was used and how regular calibrations of the instrument were performed during the four-year period. Moreover, it is unclear if the 10% uncertainty comes from the calibration uncertainty or is attributed to the uncertainties of absorption cross sections and quantum yields.

Line 175: "... close to the PKUERS site" should be specified in km.

Line 184: "This wavelength (380 nm) was chosen as it is more representative of j(NO2)" Why wasn't the AOD at 340 nm used as well to estimate AODs more representative for j(O1D) (around 300 nm), e.g. by the Angstrom equation? You can argue with better comparability with Gerasopoulos et al. , 2012 but that should be made clear.

Line 185: SSA measurements during a period of one month are hardly representative for four years. Since the AOD-SSA relationship becomes important later to explain the steep decrease of j-values with AOD, I wonder why AERONET based SSA are not consulted for the whole period.

Line 191: The source of the ozone column data should be specified and a citation included.

Line 198: "... under cloudless conditions." Was there an additional cloud screening performed or was any period marked cloud-free by AERONET taken? Because of the

distance between the sites there were certainly some cases when clouds were present at PKUERS and no clouds at the AERONET site? Moreover, to assess the importance of this work, it would be interesting to learn what fractions of daytimes were identified as clear-sky during the four years. This could be included in Table 2 for the different seasons.

Line 203: "Global irradiance" is a different quantity than actinic flux.

Line 204: Explain "AE". Were the AE taken from AERONET, was a constant AE used, or was AE set to zero to simulate with a wavelength-independent AOD? This is important later for the model measurement comparisons in Fig. 5.

Line 205: Were mean Earth-Sun distances used in the calculations or were time, date and location specified? If not, were the measured j-values scaled to a common mean Earth-Sun distance?

Line 205: Were the same absorption cross sections and quantum yields used to calculate j(O1D) and j(NO2) from TUV-derived spectra? What temperatures were used?

Line 230-234: Equations E2, E3 are not self-explaining. At least give a citation where these formulas are rationalized and explain "$\theta$".

Results and discussion

Lines 239-278: "In order to evaluate the extinction capacity..." The motivation to look into the relationship between PM2.5 and AOD should be made clearer and the results shown in Fig. 1 and Fig. 2 should be reassessed. Obviously, PM2.5 is not a good proxy to estimate AOD. Moreover, the summer-winter differences in the slopes in Fig. 1 are probably explainable by the different heights of the boundary layers alone and there is no basis to speculate about seasonal differences of aerosol optical properties unless you consult AERONET data. My conclusion of Sect. 3.1 and the first paragraph of Sect. 3.2 would be that PM2.5 is not suitable to estimate AOD (and consequently, PM2.5 data are not used in the remainder of the text). On the other hand, did you

check the relationship between PM2.5 and e.g. j(NO2) directly? I assume it looks much poorer than the relationship between j(NO2) and AOD which would confirm the assumption that AOD is a more relevant parameter.

Line 249, Table 2: Table 2 should be mentioned in Section 2.1, not here. Please consider significant digits in Tab. 2 and specify season periods in the caption.

Figs. 2-3: Specify what is shown here. Averages, medians? The periods defined as "spring", "summer", "autumn" and "winter" should be defined clearly somewhere. Are the PM2.5 data in Fig. 2b also from clear-sky days only? Specify "AOD (380 nm)" in the caption of Fig. 2.

Lines 285-292: 1) What do the stated differences in photolysis frequencies refer to? Mean daily maxima? Please specify. 2) What are the uncertainties of these differences? 3) Does the TUV-derived difference refer to aerosol-free conditions? 4) What role plays the temperature, certainly lower in Beijing during the winter compared to conditions in Crete?

Fig. 4: Specify in the caption what the full lines show. Averages, medians? What AOD bin size was used? Indicate AOD (380 nm).

Fig. 5: Add standard deviations to the measured values. Otherwise the relevance of the differences compared to the model calculations cannot be assessed. Specify the ozone column range of the measured data in the caption. Indicate AOD (380 nm) for the measured data and AOD $\neq$ f($\lambda$) for the model calculations (if that applies).

Line 319-326: Here the question again arises, what AE was used in the TUV calculations, what temperatures and if the annual changes in Sun-Earth distances were considered.

Lines 327-341 and lines 341-347: These sections are too speculative without consulting AERONET data. As already mentioned, the 1-month data in Fig. 6 is probably not representative for the average aerosol over the four year measurement period.

Lines 357-361 and Table 3: Consider significant digits.

Figure 7: What do the full lines show?

Line 397: Equation E5 should appear here.

Lines 397-406, Tables 5 and 6: 1) If ozone columns have no significant influence on j(NO2), why does Tab. 5 give four different parametrizations for four different ozone column ranges? A single parametrization should be given here to make things easier for readers who want to use these formulas. 2) What is the nature of the error limits of the parameters a1-a6 and are they of any relevance to estimate the quality of the parameterizations? Please note that for j(NO2) most parameters vary more strongly if different ozone column ranges are compared than indicated by the errors of the parameters. So these errors have no relevance and pretend an accuracy that is not real. 3) Did you systematically test if simpler parameterizations give satisfactory results as well by taking out single parameters? 4) For j(O1D) the parametrization appears arbitrary: parameters show no clear trend with ozone column although this would be expected even for an empirical formula. It would be more convincing to use a parameterization that contains SZA, AOD and ozone columns in a single formula. 5) Given that the data were probably (i) not normalized to a common Sun-Earth distance, nor (ii) to the same temperature; (iii) the AOD (380 nm) used does not apply strictly to the j(O1D) wavelength range, (iv) only 30 DU wide ranges of ozone columns were merged, and (v) cloud-sceening cannot be perfect, the obtained r2>0.95 is remarkable, also compared to Tab. 3 and 4, and should be rationalized.

Lines 407-409: What do the percentage reductions refer to? See also abstract and conclusions.

Line 411: "... and lower SSA in winter" Was not shown.

Lines 431-433: As mentioned above, this statement is not justified and the use of PM2.5 would most likely lead to no improvement of estimated j(O1D) or j(NO2) unless

you can show it directly.

Line 460, Figure 8: I assume what is shown in Fig. 8, and the 25% reduction stated in the text, refer to mean daytime ozone productions. Please specify time period.

Figure 9: How were the data shown derived, i.e. what periods of time do single data points represent?

Figure 10: Indicate in the caption that the data represent mean values over a period of one month (or n clear-sky days) in August 2012.

Figure 11: In the caption refer to Table 7 to explain the meaning of day A and day B.

Table 7, 8: Mind significant digits.

Conclusions

Conclusions 1) and the first part of 2) are speculative.

Specify meaning of percentage reductions in 3).

Line 559: Measurement campaign in August 2012 or 2014?
* * *

---

## Author Comment (AC1) · 30 Apr 2019

We greatly appreciate the time and effort that the Referees spent in reviewing our manuscript. The comments are really thoughtful and helpful to improve the quality of our paper. We have addressed each comment below. Our reply, the previous manuscript, and the revised manuscript are provided in supplement.

Please also note the supplement to this comment:
https://www.atmos-chem-phys-discuss.net/acp-2019-84/acp-2019-84-AC1-supplement.zip

---

## Author Comment (AC2) · 30 Apr 2019

We greatly appreciate the time and effort that the Referees spent in reviewing our manuscript. The comments are really thoughtful and helpful to improve the quality of our paper. We have addressed each comment below. Our reply, previous manuscript, and revised manuscript are provided in supplement.

Please also note the supplement to this comment:
https://www.atmos-chem-phys-discuss.net/acp-2019-84/acp-2019-84-AC2-supplement.zip

---

## Author Comment (AC3) · 29 May 2019

The comment was uploaded in the form of a supplement:
https://www.atmos-chem-phys-discuss.net/acp-2019-84/acp-2019-84-AC3-supplement.zip

---

## Author Comment (AC4) · 29 May 2019

https://www.atmos-chem-phys-discuss.net/acp-2019-84/acp-2019-84-AC4-supplement.zip

---

## Author Comment (AC5) · 30 May 2019

The comment was uploaded in the form of a supplement:
https://www.atmos-chem-phys-discuss.net/acp-2019-84/acp-2019-84-AC5-supplement.zip

---

## Author Comment (AC6) · 2 Jun 2019

The comment was uploaded in the form of a supplement:
https://www.atmos-chem-phys-discuss.net/acp-2019-84/acp-2019-84-AC6-supplement.zip

---

## Author Comment (AC7) · 3 Jun 2019

The comment was uploaded in the form of a supplement:
https://www.atmos-chem-phys-discuss.net/acp-2019-84/acp-2019-84-AC7-supplement.zip

---

## Author Comment (AC8) · 3 Jun 2019

The comment was uploaded in the form of a supplement:
https://www.atmos-chem-phys-discuss.net/acp-2019-84/acp-2019-84-AC8-supplement.zip

---

## Author Response (AR1)

Corresponding to mshao@pku.edu.cn.

We greatly appreciate the time and efforts that the Referees spent in reviewing our manuscript. The comments are really thoughtful and helpful to improve the quality of our paper. We have addressed each comment below, with the Referee comment in black text, our response in blue text, and relevant manuscript changes noted in red text.
* * *
**Anonymous Referee #1**

Line 30: "In addition, the slopes are equal to ...." 1) The wording: "In addition, ..." is awkward. 2) Why there is a range of slopes for $j(O^1D)$ and a single value for $j(NO2)$ is unclear at this point, and what the slopes refer to in the first place when the relationships are non-linear. 3) The slopes should be negative in any case. 4) The authors should take into account significant digits (throughout the paper). The precision of the data does not justify a statement "4.21-6.93". I would say "4.2-6.9" at the very most. 5)

AOD has to be specified here, i.e. AOD (380 nm)?

Response:1) I have removed the phrase "In addition".

2) There is a range of slopes for $j(O^1D)$ is because that the slopes varied at different total ozone column. Table 3 gives different slopes for $j(O^1D)$ at different ozone column classes. For $j(NO_2)$, total ozone column has a negligible influence on the slope. The slopes are at AOD smaller than 0.7, where the relationship between j-value and AOD is close to linear. We chose this range of AOD (AOD<0.7) to match the result in Crete by Gerasopoulos et al. (2012).

3) Yes, the slopes should be negative in any case. Therefore, in Line 30, I changed "the slopes" into "the absolute values of slopes".

4) Many thanks and I have taken into account significant digits throughout the paper.

5) I have added AOD (380 nm) in line27.

Line 27: Both j(O$^1$D) and j(NO$_2$) display significant dependence on AOD (380nm) with a nonlinear negative correlation.

Line 30-34: The absolute values of slopes are equal to 4.2-6.9•10$^{-6}$ s$^{-1}$ and 3.2•10$^{-3}$ s$^{-1}$ per AOD unit for j(O$^1$D) and j(NO$_2$) respectively at SZA of 60° and AOD smaller than 0.7, both of which are larger than those observed in a similar, previous study in the Mediterranean.
* * *
Line 32: "... larger than those observed in the Mediterranean." I would say: "... than those observed in a similar, previous study in the Mediterranean."

Response: Thank you and I have revised it in the manuscript.

Line 30-34:The absolute values of slopes are equal to 4.2-6.9•10$^{-6}$ s$^{-1}$ and 3.2•10$^{-3}$ s$^{-1}$per AOD unit for j(O$^1$D) and j(NO$_2$) respectively at SZA of 60° and AOD smaller than 0.7, both of which are larger than those observed in a similar, previous study in the Mediterranean.
* * *
Line 33: "...have a stronger extinction on ..." Please reword.

Response: I have changed it into "...have a stronger extinction effect on ...".

Line 34-36: This indicates that the aerosols in urban Beijing have a stronger extinction effect on actinic flux than absorptive dust aerosols in the Mediterranean.
* * *
Line 38, 39: "... j(NO2) by 24.2% and 30.4% for summer and winter, ... j(O1D) by 27.3% and 32.6%..." 1) The meaning of these numbers is unclear. I assume they refer to some kind of seasonal mean of the photolysis frequencies that needs to be specified. 2) The precision implied by three digits is misleading.

Response: Thank you and I have revised it. We calculated the reduction ratios of j values in the following procedure: We use the parametric equations (Table 5 and Table

6) to calculated J(O1D) and J(NO2) using corresponding SZA, AOD and ozone column at corresponding time (5 minute average). Two situations are calculated: One, AOD is equal to 0 at all times. Two, AOD is equal to the observed values at all times. The mean values of J(O1D) and J(NO2) for summer and winter are calculated in the two situations and the reduction ratios can be calculated accordingly.

Line 39-42: According to the parametric equation, aerosols lead to a decrease in seasonal mean $j(NO_2)$ by 24% and 30% for summer and winter, respectively, and the corresponding decrease in seasonal mean $j(O^1D)$ by 27% and 33% respectively, compared to an aerosol-free atmosphere (AOD = 0).
* * *
Line 42: "... the monthly average net ozone production is reduced by 25%." By looking at Fig. 10, I assume the 25% refers to a monthly mean daytime net ozone production that needs to be specified.

Response: Yes, the 25% refers to a monthly mean daytime net ozone production. I have specified it.

Line 44: The simulation results shows that the monthly mean daytime net ozone production rate is reduced by up to 25% due to the light extinction of aerosols.
* * *
Lines 54, 59, 63, 65, 66: Use consistent notations for O(3P) and O(1D).

Response: Thank you and I have revised it.
* * *
Line 60: "....the only significant chemical source...."

Response: Thank you and I have revised it.

Line 63-64: which is the only significant chemical source of ozone in the troposphere (Finlayson-Pitts et al., 2000).
* * *
Lines 68-71: Remove the symbol "S" in the brackets. It may be added as an index to "sigma" and "phi" but "S" is no variable like "lambda" or "T".

Response: Thank you and I have revised it.
* * *
Lines 78-79: "Scattering aerosols can enhance..., while absorptive aerosols reduce ...throughout the boundary layer." These statements are unclear and certainly do not apply for all conditions.

Response: I agree with you that these statements are unclear and certainly do not apply for all conditions. Therefore, I add "Some previous studies showed that" in front of this sentence. Both the abstract of Dickerson et al. (1997) and the introduction of Flynn et al. (2010) have put forward this viewpoint.

Reference:

Dickerson, R. R., Kondragunta, S., Stenchikov, G., Civerolo, K. L., Doddridge, B. G., Holben, N.: The impact of aerosols on solar ultraviolet radiation and photochemical smog, Science, 278, 827–830, 10.1126/science.278.5339.827, 1997.

Flynn, J., Lefer, B., Rappenglück, B., Leuchner, M., Perna, R., Dibb, J., Ziemba, L., Anderson, C., Stutz, J., Brune, W., Ren, X. R.: Impact of clouds and aerosols on ozone production in Southeast Texas. ATMOSPHERIC ENVIRONMENT, 44, 4126–4133, 10.1016/j.atmosenv.2009.09.005, 2010.

Line 85-88: Previous studies showed that scattering aerosols can enhance the actinic flux throughout the troposphere, while absorptive aerosols reduce the actinic flux throughout the boundary layer (Jacobson, 1998; Dickerson et al., 1997; Castro et al., 2001; Flynn et al., 2010).
* * *
Line 91: "Therefore it is necessary to quantitatively evaluate the effect of aerosols on photolysis frequencies for the purpose of ozone prevention". I would say: "... for a better understanding of ozone formation under highly polluted conditions."

Response: Thank you and I have revised it.

Line 98: Therefore, it is necessary to quantitatively evaluate the effect of aerosols on photolysis frequencies for a better understanding of ozone formation under highly polluted conditions.

- - - - - - - - - - - - - - - - - - - - - - - - - - - - - - - - - - - - - - - - - - - -

Line 98: "... are compared with the observed value to test the simulation effect." Unclear: If radiative transfer models are used there are usually no measurements available. And what is the "simulation effect"?

Response: The expressed meaning is unclear and thus I simplified this sentence.

Line 105: The observed data of related influential factors of the photolysis frequencies are taken as the model's input to calculate the photolysis frequencies.

- - - - - - - - - - - - - - - - - - - - - - - - - - - - - - - - - - - - - - - - - - - -

Line 100: "... due to complicated environmental conditions...". Clarify.

Response: The complicated environmental conditions will influence the simulation of photolysis frequencies by radiative transfer model. Although AOD is easily acquired from ARONET and satellite, SSA, asymmetry factor, AE and the vertical profile of aerosols are not always available, and their observed values often had a large uncertainty. The complicated environmental conditions including relative humidity, temperature, planetary boundary layer height and emission characteristics all influence aerosol optical properties and thus influence light extinction of aerosols. Previous studies indicated that the aging of black carbon, mixing state of aerosols, the absorptive capacity of organic aerosols and the hygroscopicity of aerosols, all of which are determined by specific environmental condition, significantly contribute to the uncertainty of aerosol optical properties and thus influence of the simulation of photolysis frequencies by radiative transfer model (Liao et al., 1999; Moffet et al., 2009; Gyawali et al., 2009; Lack and Cappa, 2010; Barnard et al., 2008; Jeong and Sokolik., 2007).

Reference:

Liao, H., Yung, Y. L., and Seinfeld, J. H.: Effects of aerosols on tropospheric photolysis rates in clear and cloudy atmospheres, JOURNAL OF GEOPHYSICAL RESEARCH, 104(D19), 23697–23707, 1999.

Moffet, R. C. and Prather, K.: In-situ measurements of the mixing state and optical properties of soot with implications for radiative forcing estimates, P. Natl. Acad. Sci., 106(29), 11872–11877, 2009.

Gyawali, M., Arnott, W. P., Lewis, K., and Moosmüller, H.: Insitu aerosol optics in Reno, NV, USA during and after the summer 2008 California wildfires and the influence of absorbing and non-absorbing organic coatings on spectral light absorption, Atmos. Chem. Phys., 9, 8007–8015, doi:10.5194/acp-9-8007-2009, 2009.

Lack, D. A. and Cappa, C. D.: Impact of brown and clear carbon on light absorption enhancement, single scatter albedo and absorption wavelength dependence of black carbon, Atmos. Chem. Phys., 10, 4207–4220, doi:10.5194/acp-10-4207-2010, 2010.

Barnard, J. C., Volkamer, R., and Kassianov, E. I.: Estimation of the mass absorption cross section of the organic carbon component of aerosols in the Mexico City Metropolitan Area, Atmos. Chem. Phys., 8, 6665–6679, doi:10.5194/acp-8-6665-2008, 2008.

Jeong, G. R; Sokolik, I. N., Effect of mineral dust aerosols on the photolysis rates in the clean and polluted marine environments. JOURNAL OF GEOPHYSICAL RESEARCH-ATMOSPHERES, 112, D21, 2007.

- - - - - - - - - - - - - - - - - - - - - - - - - - - - - - - - - - - - - - - - - - - - - - -

Line 123-136: Use the term "photolysis frequencies" consistently throughout the text.

Response: Thank you and I have revised it.
* * *
Lines 134, 135: Mind significant digits, see abstract by Li et al., 2011.

Response: Thank you and I have revised it.

Line 142-146: This study showed that the daily average $j(O^1D)$ in the troposphere at the altitude of 1 km, 3 km, and 10 km from the ground was reduced by 53%, 37%, and 21%, respectively, resulting in a decrease in the ozone concentration by 5.4%, 3.8%, and 0.10% in the three layers.
* * *
Line 143ff: "Our overall goal..." It should be made clear that this study was strongly informed by a similar work by Gerasopoulos et al., 2012 which is not adequately referred to in the Introduction.

Response: Thank you and I have added this part.

Line 157-159: The relationship between AOD and photolysis frequencies is adequately compared with previous study in the Mediterranean (Casasanta et al., 2011; Gerasopoulos et al., 2012).
* * *
Line 155: The exact measurement period should be given here. Was it exactly four years?

Response: I have described the exact measurement period.

Line 161: From August 2012 to December 2015, $j(O^1D)$ and $j(NO_2)$ were measured continuously at PKUERS site. The data of the period during October 2012 to March 2013 and August 2015 are missed due to instrument maintenance and other measurement campaigns.
* * *
Line 163: What absorption cross sections and quantum yields were used to calculate the photolysis frequencies?

Response: The quantum yields of $J(O^1D)$ was taken from Matsumi et al.(2002), while the ozone cross section was derived from Daumont et al. (1992) and Malicet et al. (1995). For $j(NO_2)$, the quantum yields used was taken from Bass et al. (1976) and Davenport et al. (1978), while the cross section was derived from Jones and Bayes (1973), Harker et al. (1977) and Davenport (1978). I have added these sentences in the Manuscript.

Line 186-191: For $j(O^1D)$, the quantum yield used was taken from Matsumi et al.(2002), while the absorption cross section was derived from Daumont et al. (1992) and Malicet et al. (1995). Measured temperature was used to retrieve ozone absorption cross section and quantum yield. For $j(NO_2)$, the quantum yields used was taken from Bass et al. (1976) and Davenport et al. (1978), while the absorption cross section was derived from Jones and Bayes (1973), Harker et al. (1977) and Davenport (1978).
* * *
Line 163: I assume the $j(O1D)$ were calculated temperature dependent according to Eq. 1 (and a statement in lines 302-304). That should be clearly stated. However, is it useful, if a common parameterization as a function of AOD is later used for summer and winter $j(O1D)$? There may be arguments to include temperature but the influence should be mentioned and quantified in Section 3 (see below).

Response: The $j(O1D)$ were calculated temperature dependent. I added the sentence "Measured temperature are used to retrieve ozone absorption cross section and photodissociation quantum yield." in line 188. I have evaluated the impact of temperature on $j(O^1D)$ by calculating the ratio of $j(O^1D)$ to $j(O^1D)$ at temperature=298K (Figure S2). The result indicates that temperature changed $j(O^1D)$ by no more than 20%. In addition, the determination coefficients of fitted parametric equations are larger than 0.95, indicating the influence of temperature is relatively small.

Line 458-469: For $j(O^1D)$, both of $O_3$ column and temperature affect $j(O^1D)$ significantly. Figure S1 presents the dependence of $j(O^1D)$ on ozone column at low AOD level (AOD<0.3) and SZA of (a) 30°±1° and (b) 60°±1°, respectively. Ozone column ranging from 270 to 400 DU leads to $j(O^1D)$ reducing about 50%. In order to evaluate the impact of temperature on $j(O^1D)$, we calculated the ratio of $j(O^1D)$ at measured temperature to $j(O^1D)$ at temperature = 298K ($j(O^1D)/ j(O^1D)_{T=298K}$) (Figure S2). $j(O^1D)/j(O^1D)_{T=298K}$ varied from 0.82 to 1.03 indicating that temperature changed $j(O^1D)$ by no more than 20%. Therefore, temperature played a minor role in changing $j(O^1D)$ compared with ozone column. As a result, when we fitted the relationship among $j(O^1D)$, AOD and cos(SZA), the effect of ozone column is considered but the effect of temperature is not considered.

[Figure]

Figure S1. Dependence of $j(O^1D)$ on AOD (380nm) at low AOD level (AOD<0.3) and SZA of (a) 30°±1° and (b) 60°±1°, respectively.

[Figure]

Figure S2. The time series of the monthly mean ratio of $j(O^1D)$ to $j(O^1D)_{T=298K}$ ($j(O^1D)/ j(O^1D)_{T=298K}$) from August 2012 to December 2015.
* * *
Line 164: "Shetter and Müller, 1999"

Response: Thank you and I have revised it.
* * *
Line 169: The studies by Shetter and Müller, 1999, and Hofzumahaus et al., 1999 describe double-monochromator based instruments with somewhat different properties.

The authors should state what type of detector was used and how regular calibrations of the instrument were performed during the four-year period. Moreover, it is unclear if the 10% uncertainty comes from the calibration uncertainty or is attributed to the uncertainties of absorption cross sections and quantum yields.

Response: (1) The double-monochromators for wavelength separation and successive measurements with single detectors (e.g. photomultipliers) upon scanning the wavelength. This is excellent for stray light suppression which is important in the

UV-B range (e.g. Shetter and Müller, 1999; Hofzumahaus et al., 1999). Drawbacks are the comparatively long time periods to complete the wavelength scans (≥30s) and the use of motor-driven optical components which may cause stability problems under field measurement conditions. Our method uses single monochromators and detector arrays (e.g. photodiode arrays) for simultaneous measurements covering the whole range of relevant wavelengths. This method has the advantage of high time-resolution and stability because no movable parts are involved. (2) The detector is a 2048×64 pixels photodiode array detector. (3) The 10% uncertainty is associated with the quartz receiver and stray-light effects.

Line177-194: The actinic flux was measured using a spectroradiometer and the photolysis frequencies were calculated from the absorption cross section and quantum yield of each species (Shetter and Müller, 1999). The spectroradiometer consisted of a single monochromator with a fixed grating (CARL ZEISS), an entrance optic with a $2\pi$ steradian (sr) solid angle quartz diffusor and a 2048×64-pixel photodiode array detector. The spectral measurements were performed with a wavelength resolution of 2 nm, covering a wavelength range of 290-650 nm (Hofzumahaus et al., 1999). A 1000 W National Institute of Standard and Technology (NIST) traceable lamp was used for calibration under laboratory conditions (Bohn et al., 2008). The measured spectra were corrected for dark signal and stray light. For $j(O^1D)$, the quantum yield was taken from Matsumi et al.(2002), while the ozone cross section was derived from Daumont et al. (1992) and Malicet et al. (1995). Measured temperature was used to retrieve ozone absorption cross section and quantum yield. For $j(NO_2)$, the quantum yield was taken from Bass et al. (1976) and Davenport et al. (1978), while the cross section was derived from Jones and Bayes (1973), Harker et al. (1977) and Davenport (1978). The calculated photolysis frequencies had a time resolution of 10 s and an accuracy of ±10% including uncertainties associated with the quartz receiver and stray-light effects.

- - - - - - - - - - - - - - - - - - - - - - - - - - - - - - - - - - - - - - - - - - - - - - - -

Line 198: "... under cloudless conditions." Was there an additional cloud screening performed or was any period marked cloud-free by AERONET taken? Because of the distance between the sites there were certainly some cases when clouds were present at PKUERS and no clouds at the AERONET site? Moreover, to assess the importance of this work, it would be interesting to learn what fractions of daytimes were identified as clear-sky during the four years. This could be included in Table 2 for the different seasons.

Response: (1) The cloudless conditions are identified according to the presence of AOD data in AERONET since AOD data is unavailable under cloudy conditions. However, we didn't have additional cloud screening procedure. (2) I agree with you that the distance between the sites will cause some cases when clouds were present at PKUERS and no clouds at the AERONET site, which may disturb our analysis of the relationship between AOD and photolysis frequencies. Due to the close distance between the two sites (6.4km), this influence maybe relatively small. (3) Many thanks.

I added daytime clear-sky fraction for the different seasons in Table 2.

Line 208-210: The daytime clear-sky conditions were identified according to the presence of AOD data of AERONET since AOD data are unavailable under cloudy conditions.

Line 219-221: Table 1 presents $O_3$ column concentration, temperature, relative humidity, daytime clear-sky fraction and respective standard deviation for different seasons.

Line 1019:

Table 1. $O_3$ column concentration, temperature, relative humidity, daytime clear-sky fraction and respective standard deviation for different seasons.

| Season | $O_3$ column (Du) | Temperature (℃) | Relative humidity (%) | Clear-sky fraction(%) |
|--------|-------------------|-----------------|-----------------------|-----------------------|
| Spring | 355±37 | 16±7.8 | 33±18 | 41 |
| Summer | 310±24 | 28±4.2 | 57±18 | 36 |
| Autumn | 304±23 | 16±7.4 | 46±21 | 42 |
| Winter | 347±28 | 0.53±4.2 | 30±18 | 41 |

- - - - - - - - - - - - - - - - - - - - - - - - - - - - - - - - - - - - - - - - - - - - - - -

Line 203: "Global irradiance" is a different quantity than actinic flux.

Response: Thank you and I have changed "Global irradiance" into "actinic flux spectra".

Line 231-234: In order to solve the radiative transfer equation, TUV uses the discrete-ordinates algorithm (DISORT) with 4 streams and calculates **the actinic flux spectra** with wavelength range of 280-420 nm in 1 nm steps and resolution.

- - - - - - - - - - - - - - - - - - - - - - - - - - - - - - - - - - - - - - - - - - - - - - -

Line 204: Explain "AE". Were the AE taken from AERONET, was a constant AE used, or was AE set to zero to simulate with a wavelength-independent AOD? This is important later for the model measurement comparisons in Fig. 5.

Response: AE (380/550nm) are taken from AERONET and the mean value of 1.3 during August 2012 - December 2015 is used in TUV model to simulate the wavelength dependency of AOD.

Line 210-211: AE were also acquired from ARONET.

Line 239-240: AE(380/550nm) is taken from AERONET and the mean value of 1.3 during June 2012 - December 2015 is used in TUV model.
* * *
Line 205: Were mean Earth-Sun distances used in the calculations or were time, date and location specified? If not, were the measured j-values scaled to a common mean Earth-Sun distance?

Response: Mean Earth-Sun distances was used in the calculations of TUV model and measured j-values were scaled to the mean Earth-Sun distance.

Line 377-381: The observed $j(O^1D)$ was at ozone column of 330-360DU and were scaled to the temperature of 298K. AE(380/550nm) = 1.3, ozone column = 345 and Temperature = 298K were used in TUV model for all simulations. Mean Earth-Sun distance was used in the calculations of TUV model and measured j-values were scaled to the mean Earth-Sun distance.
* * *
Line 205: Were the same absorption cross sections and quantum yields used to calculate j(O1D) and j(NO2) from TUV-derived spectra? What temperatures were used?

Response: Measured temperatures were used to calculate the absorption cross sections and quantum yields of $j(O^1D)$. The absorption cross sections and quantum yields of j(NO2) at 298K were used since they are influenced negligibly by temperature.
* * *
Line 230-234: Equations E2, E3 are not self-explaining. At least give a citation where these formulas are rationalized and explain "θ".

Response: Many thanks. I have gave a citation and explained "θ".

Line 258-261: Net ozone production is equal to the reaction rate between peroxy radicals (RO$_2$ and HO$_2$) and NO minus the loss rate of NO$_2$ and O$_3$ as shown in E2, E3, and E4 a**s derived by Mihelcic et al. (2003)**.

Line 269-270: where θ is the fraction of O$^1$D from ozone photolysis that reacts with water vapor.
* * *
Lines 239-278: "In order to evaluate the extinction capacity..." The motivation to look into the relationship between PM2.5 and AOD should be made clearer and the results shown in Fig. 1 and Fig. 2 should be reassessed. Obviously, PM2.5 is not a good proxy to estimate AOD. Moreover, the summer-winter differences in the slopes in Fig. 1 are probably explainable by the different heights of the boundary layers alone and there is no basis to speculate about seasonal differences of aerosol optical properties unless you consult AERONET data. My conclusion of Sect. 3.1 and the first paragraph of Sect. 3.2 would be that PM2.5 is not suitable to estimate AOD (and consequently, PM2.5 data are not used in the remainder of the text). On the other hand, did you check the relationship between PM2.5 and e.g. j(NO2) directly? I assume it looks much poorer than the relationship between j(NO2) and AOD which would confirm the assumption that AOD is a more relevant parameter.

Response: (1) I have revised the motivation to look into the relationship between PM2.5 and AOD. Compared with AOD, PM$_{2.5}$ is a more common proxy to evaluate the level of particulate matter pollution in spite that AOD is a more closely related parameter of photolysis frequencies. As a result, we attempted to analyze the quantitative relationship between PM$_{2.5}$ and AOD to evaluate the influence of PM$_{2.5}$ on AOD and thus on photolysis frequencies. (2) I agree with you that PM$_{2.5}$ is not a good proxy to estimate AOD due to multiple interference factors including relative humidity, planetary boundary layer height, aerosol type, aerosol size distribution, aerosol distribution in the vertical direction. Zheng et al. (2017) studied the influential factors for the relationship between PM2.5 and AOD in Beijing during 2011-2015. He found that in addition to RH and PBLH, aerosol components (scattering or absorptive) and size (coarse mode or fine mode) also influenced the slope of PM2.5 vs AOD significantly according to SSA, AE and FMF data of AERONET. Since Zheng et al. (2017) have studied this question in detail in the same region and the same period, I think that our work needn't analyze SSA, AE and FMF data of AERONET again. Instead, we just compared the slope of PM2.5 vs AOD of our study with other study in Beijing and other cities of North China. I agree with you that the conclusion is that PM2.5 is not suitable to estimate AOD due to the large uncertainty. (3) I have checked the relationship between PM2.5 and j(NO2) directly, it is true that there is much poorer correlation than the relationship between j(NO2) and AOD.

Line 276-281: Compared with AOD, $PM_{2.5}$ is a more common proxy to evaluate the level of particulate matter pollution in spite that AOD is a more closely related parameter of photolysis frequencies. As a result, we attempted to analyze the quantitative relationship between $PM_{2.5}$ and AOD to evaluate the influence of $PM_{2.5}$ on AOD and thus on photolysis frequencies.

Line 307-308: Consequently, using $PM_{2.5}$ to estimate AOD has a large uncertainty due to multiple interference factors.

- - - - - - - - - - - - - - - - - - - - - - - - - - - - - - - - - - - - - - - - - - - - - - - - - - - - - - -

Line 249, Table 2: Table 2 should be mentioned in Section 2.1, not here. Please consider significant digits in Tab. 2 and specify season periods in the caption.

Response: Many thanks, and I have revised it. I change Table 2 into Table 1.

Line 218-222: In addition, meteorological parameters such as temperature, relative humidity, and pressure were simultaneously observed at this site. Table 1 presents total $O_3$ column, temperature, relative humidity, daytime clear-sky fraction and respective standard deviation for different seasons.

Line 1001-1004:

Table 1. $O_3$ column concentration, temperature, relative humidity, daytime clear-sky fraction and respective standard deviation for different seasons (spring: March, April and May; summer: June, July and August; autumn: September, October and November; winter: December, January and February).

| Season | $O_3$ column (Du) | Temperature (℃) | Relative humidity (%) | Clear-sky fraction(%) |
|---|---|---|---|---|
| Spring | 355±37 | 16±7.8 | 33±18 | 41 |
| Summer | 310±24 | 28±4.2 | 57±18 | 36 |
| Autumn | 304±23 | 16±7.4 | 46±21 | 42 |
| Winter | 347±28 | 0.53±4.2 | 30±18 | 41 |
* * *
Figs. 2-3: Specify what is shown here. Averages, medians? The periods defined as "spring", "summer", "autumn" and "winter" should be defined clearly somewhere. Are
the PM2.5 data in Fig. 2b also from clear-sky days only? Specify "AOD (380 nm)" in the caption of Fig. 2.

Response: (1) Figs. 2-3 show mean values. I have specified it. (2) Thank you, I have define "spring", "summer", "autumn" and "winter" in Line 285-287. (3) The previous PM2.5 data in figure 2b are from all-sky days. I have revised Figure 2b using PM2.5 data from clear-sky days.

Line 1133: Figure 2. Diurnal cycles of (a) mean AOD and (b) mean PM$_{2.5}$ in the four seasons under cloudless conditions.

Line 1152: Figure 3. Diurnal cycles of (a) mean j(O$^1$D) and (b) mean j(NO$_2$) in the four seasons under cloudless conditions.

Line 284-287: The determination coefficient ($r^2$) is 0.53, 0.58, 0.62 and 0.59 for spring (March, April and May), summer (June, July and August), autumn (September, October and November) and winter (December, January and February), respectively.
* * *
Lines 285-292: 1) What do the stated differences in photolysis frequencies refer to? Mean daily maxima? Please specify. 2) What are the uncertainties of these differences? 3) Does the TUV-derived difference refer to aerosol-free conditions? 4) What role plays the temperature, certainly lower in Beijing during the winter compared to conditions in Crete?

Response: 1) The stated differences in photolysis frequencies refer to mean daily maxima and I have specified it in the manuscript. 2) I have added the uncertainties of these differences. 3) Yes, the TUV-derived difference refers to aerosol-free conditions and I have specified it in the manuscript. 4) I have added the analysis of the influence of temperature during winter.

Line 326-330: The observed mean daily maxima of photolysis frequencies at this site are lower than that observed in the eastern Mediterranean (Crete, Greece, 35°20′N,25°40′E) (Gerasopoulos et al., 2012) by $7.8\times10^{-6} \pm 5.5\times10^{-6}$ s$^{-1}$ and $4.9\times10^{-6} \pm 1.8\times10^{-6}$ s$^{-1}$ for $j(O^1D)$, and $1.9\times10^{-3} \pm 1.2\times10^{-3}$ s$^{-1}$ and $3.3\times10^{-3} \pm 1.0\times10^{-3}$ s$^{-1}$ for $j(NO_2)$, in summer and winter respectively.

Line 330-334: The corresponding lower photolysis frequencies of Beijing than the eastern Mediterranean due to SZA difference is $1.7\times10^{-6}$ s$^{-1}$ and $3.0\times10^{-6}$ s$^{-1}$ for $j(O^1D)$, and $8.0\times10^{-5}$ s$^{-1}$ and $6.6\times10^{-4}$ s$^{-1}$ for $j(NO_2)$ according to TUV model under aerosol-free conditions, which are significantly lower than observed decreased magnitudes.

Line 334-339: Additionally, we know that the temperature is lower in Beijing during the winter compared to conditions in Crete. The measured mean temperature in Beijing during winter is equal to 0.53±4.2 °C (Table 1). When we consider the mean temperature in Crete (about 10 °C) is 10 °C higher than in Beijing, the lower $j(O^1D)$ of Beijing than Crete is $5.5 \times 10^{-7}$ s$^{-1}$, which is also not able to compensate the $j(O^1D)$ gap between the two regions during winter.
* * *
Fig. 4: Specify in the caption what the full lines show. Averages, medians? What AOD bin size was used? Indicate AOD (380 nm).

Response: The full lines are fitted by exponential function. The coefficients of determination ($r^2$) vary from 0.5 to 0.8. I have specified it in fig. 4.

Line 1168-1170: Figure 4. Dependence of $j(O^1D)$ on AOD (380nm) at SZA of (a) 30° and (b) 60° and at different classes of ozone column concentration: 300-330 DU (red), 330-360 DU (blue), and 360-390 DU (green). The full lines are fitted by exponential function.
* * *
Fig. 5: Add standard deviations to the measured values. Otherwise the relevance of the differences compared to the model calculations cannot be assessed. Specify the ozone column range of the measured data in the caption. Indicate AOD (380 nm) for the measured data and AOD $\neq f(\lambda)$ for the model calculations (if that applies).

Response: (1) I have add standard deviations to the measured values. (2) The ozone column range for $j(O^1D)$ is 330-360 DU. I have specified it.
* * *
Line 319-326: Here the question again arises, what AE was used in the TUV calculations, what temperatures and if the annual changes in Sun-Earth distances were considered.

Response: We used the mean AE=1.3, temperature = 298K in TUV. In addition, we used the mean Sun-Earth distance in TUV.

Line 377-381: The observed $j(O^1D)$ is at temperature of 288-308K and ozone column of 330-360DU. AE(380/550nm) = 1.3, ozone column = 345 and Temperature = 298K are used in TUV model for all simulations. Mean Earth-Sun distance was used in the calculations of TUV model and measured j-values were scaled to the mean Earth-Sun distance.
* * *
Lines 327-341 and lines 341-347: These sections are too speculative without consulting AERONET data. As already mentioned, the 1-month data in Fig. 6 is probably not representative for the average aerosol over the four year measurement period.

Response: As mentioned above, there is a slight positive correlation between AOD and AERONET based SSA during 2012-2015. As the AERONET based SSA data have a large uncertainty, we are not sure if the result should be added into the manuscript.
* * *
Lines 357-361 and Table 3: Consider significant digits.

Response: I have revised it.
* * *
Figure 7: What do the full lines show?

Response: The full lines are fitted by exponential function.

Line 1221-1223: Figure 7. Dependence of $j(NO_2)$ on AOD (380nm) at different SZA

classes. The classes of cos(SZA) are 0–0.2 (black), 0.2–0.4 (purple), 0.4–0.6 (green),

0.6–0.8 (blue), and 0.8–1 (red). The full lines are fitted by exponential function.
* * *
Line 397: Equation E5 should appear here.

Response: Thank you and I have revised it.
* * *
Lines 397-406, Tables 5 and 6: 1) If ozone columns have no significant influence on j(NO2), why does Tab. 5 give four different parametrizations for four different ozone column ranges? A single parametrization should be given here to make things easier for readers who want to use these formulas. 2) What is the nature of the error limits of the parameters a1-a6 and are they of any relevance to estimate the quality of the parameterizations? Please note that for j(NO2) most parameters vary more strongly if different ozone column ranges are compared than indicated by the errors of the parameters. So these errors have no relevance and pretend an accuracy that is not real. 3) Did you systematically test if simpler parameterizations give satisfactory results as well by taking out single parameters? 4) For j(O1D) the parametrization appears arbitrary: parameters show no clear trend with ozone column although this would be expected even for an empirical formula. It would be more convincing to use a parameterization that contains SZA, AOD and ozone columns in a single formula. 5) Given that the data were probably (i) not normalized to a common Sun-Earth distance, nor (ii) to the same temperature; (iii) the AOD (380 nm) used does not apply strictly to the j(O1D) wave-length range, (iv) only 30 DU wide ranges of ozone columns were merged, and (v) cloud-sceening cannot be perfect, the obtained r2>0.95 is remarkable, also compared to Tab. 3 and 4, and should be rationalized.

Response: 1) I agree with you that a single parametrization should be given for $j(NO_2)$. I have revised it. 2) The error limits refer to 95% confidence bounds of these parameters. When we used a single parametric equation for all data of $j(NO_2)$, the error limits get lower significantly than that of different ozone column classes. 3) Yes,

I try using different functions to fit the relationship, and it seems that the quadratic polynomial form gave the best fit, which reflects the nonlinear relationship between AOD and j-values and considers the combined effect of the AOD and SZA on j-values. 4) I agree with you that it would be more convincing to use a parameterization that contains SZA, AOD and ozone columns in a single formula. I have revised it in this part. 5) All of the problems of the data you have mentioned is true. Even if there are these problems in the data, I acquired a remarkable r2 (r2>0.95), indicating the fitted results is relatively rationalized.

Line471:

$$j(NO_2) = a_1 + a_2 AOD + a_3 \cos(SZA) + a_4 (AOD)^2 + a_5 AOD\cos(SZA) + a_6 (\cos(SZA))^2$$

………………………E5

Line 464-470: By fitting the relationship at different ozone classes (classification width=30DU), we found that ozone column increasing by 30DU results in $j(O^1D)$ at a constant SZA and AOD decreasing by 18%. Therefore, the parametric equation for $j(O^1D)$ is transformed into the form E6, which reflects the influence of ozone column. The parameters $a_1$-$a_6$ correspond to ozone column range = 300-330 DU, thus we use 315 DU as the weighted standard of ozone column. The fitting parameters $a_1$-$a_6$ for $j(O^1D)$ is shown in Table 6.

$$j(O^1D) = [a_1 + a_2 AOD + a_3 \cos(SZA) + a_4 (AOD)^2 + a_5 AOD\cos(SZA) + a_6 (\cos(SZA))^2]$$
$$\times [1+(315 - O_3 \text{ column}) \times 0.006]$$

……………E6

- - - - - - - - - - - - - - - - - - - - - - - - - - - - - - - - - - - - - - - - - - - - - - - - - - -

Lines 407-409: What do the percentage reductions refer to? See also abstract and conclusions.

Response: The percentage reductions refer to seasonal mean values under clear-sky conditions. I have revised it.

Line 485-490: The parametric equations can be used to quantitatively evaluate the effect of AOD on photolysis frequencies in Beijing. According to the parametric equations, aerosols lead to a decrease in seasonal mean $j(NO_2)$ by 24% and 30% and a decrease in seasonal mean $j(O^1D)$ by 27% and 33% in summer and winter under clear-sky conditions, respectively, compared to an aerosol-free atmosphere.
* * *
Line 411: "... and lower SSA in winter" Was not shown.

Response: I have removed "... and lower SSA in winter".
* * *
Lines 431-433: As mentioned above, this statement is not justified and the use of PM2.5 would most likely lead to no improvement of estimated j(O1D) or j(NO2) unless you can show it directly.

Response: Yes, the use of PM2.5 leads to no improvement of estimated j(O1D) or j(NO2). Therefore, I have removed this part.
* * *
Line 460, Figure 8: I assume what is shown in Fig. 8, and the 25% reduction stated in the text, refer to mean daytime ozone productions. Please specify time period.

Response: Yes, it refer to mean daytime ozone productions and I have revised it.

Line 541-543: Figure 8. Since the decreasing amplitude of the daytime ozone production rate is far larger than that of the daytime ozone loss rate, the mean daytime net production rate of ozone is reduced by 25%.

Line 1237: Figure 8. Mean daytime ozone production and loss terms in August 2012.
* * *
Figure 9: How were the data shown derived, i.e. what periods of time do single data points represent?

Response: Single data point represent daytime hourly mean value.

Line 1250-1251: Figure 9. Correlation between $P(O_3)_{j\_obs}/P(O_3)_{j\_AOD=0}$ (or $D(O_3)_{j\_obs}/D(O_3)_{j\_AOD=0}$) and JIF of $j(NO_2)$. Single data point represent daytime hourly mean value.
* * *
Figure 10: Indicate in the caption that the data represent mean values over a period of one month (or n clear-sky days) in August 2012.

Response: I have specified it in the caption.

Line 1267-1268: Figure 10. Diurnal profiles of mean $P(O_3)_{j\_obs}$, $P(O_3)_{j\_AOD=0}$, $D(O_3)_{j\_obs}$, and $D(O_3)_{j\_AOD=0}$ in August 2012 under clear-sky conditions.

- - - - - - - - - - - - - - - - - - - - - - - - - - - - - - - - - - - - - - - - - - - - - - - - - - - - - - - -

Figure 11: In the caption refer to Table 7 to explain the meaning of day A and day B.

Response: I have specified it in the caption.

Line 1281-1286: Figure 11. Diurnal profile of net $P(O_3)$ simulated by the box model. Three cases are displayed: (1) A day (red circles): August 21, 2012 with low AOD level and high photolysis frequencies; (2) B day (blue circles): August 26, 2012 with high AOD level and low photolysis frequencies; and (3) the photolysis frequencies of B day adjusted to the level of A day with other conditions unchanged (green circles). The specific conditions of A day and B day are listed in Table 7.
- - - - - - - - - - - - - - - - - - - - - - - - - - - - - - - - - - - - - - - - - - - - - - - - - - - - - - - -

Table 7, 8: Mind significant digits.

Response: I have revised it.

**Anonymous Referee #2**

1) In the introduction, after the definition of actinic flux (line 68) the authors could include that since the photolysis rates are proportional to the actinic flux and not all stations acquire a 2π spectroradiometer or chemical actinometers for J measurements, several methods have been developed to determine actinic flux and photolysis rates from ground based measurements of irradiance (Kylling et al 2003, Kazadzis et al. 2000,

2004, Topaloglou et al. 2005, Trebs et al. 2009).

Response: Thank you and I have added this sentence in the manuscript.

Line 69-74: Since the photolysis rates are proportional to the actinic flux and not all stations acquire a 2π spectroradiometer or chemical actinometers for J measurements, several methods have been developed to determine actinic flux and photolysis frequencies from ground based measurements of irradiance (Kylling et al 2003, Kazadzis et al. 2000, 2004, Topaloglou et al. 2005, Trebs et al. 2009).
* * *
2) It is stated, in the abstract, that the reduction of J(O1D) and J(NO2) is in the order of 24.2% and 30.4% (for summer and winter respectively) while for the J(NO2) in the order of (27.3 an 32.6%) compared to an aerosol free atmosphere (aod=0?). Since the parametric equations include sza and AOD, the authors could clarifiy how exactly these percentages have been calculated i) to what sza are these percentages referring to? Also for what ozone class for J(O1D)? ii) are these maximum reductions for maximum aod observed or for a mean aod value (i.e. 0.76 )? iii) Through which parameters are summer and winter percentages calculated?

Response: Aerosol free atmosphere refers to AOD=0, and I specified it in the manuscript. We use the parametric equations (Table 5 and Table 6) to calculated

J(O1D) and J(NO2) using corresponding SZA and AOD at corresponding time (5 minute average). Two situations are calculated: One, AOD is equal to 0 at all times. Two, AOD is equal to the observed values at all times. For the calculation in the two situations, the corresponding parametric equation at different ozone classes is used according to observed ozone column at different times. The mean values of J(O1D) and J(NO2) for summer and winter are calculated in the two situations and the reduction ratio can be calculated accordingly.
* * *
3) How do the authors comment the (low) r2 coefficient in the linear fits of J(O1D) and J(NO2) versus aod for aod<0.7?

Response: For j(NO2), the relatively large SZ classification width (0.2) is the main cause of the low r2. If we shrink SZ classification width into 0.05, the r2 coefficient will be higher than 0.6. For j(O1D), the relatively large ozone column classification width (30DU) contributes to the low r2 to a large extent. In addition, the nonlinear relationship between j-values and AOD also leads to the low r2 for AOD<0.7.
* * *
4) Concerning the TUV radiation model, information (apart from ssa values) about the input that was used could be included, such as solar spectrum used, aerosol profile etc.

In p.2.2 it is stated that global irradiance spectra are calculated. Do you maybe mean actinic flux spectra? Since photolysis rates are proportional to actinic flux, has any comparison been done between the actinic flux measured by the spectroradiometer and that from the TUV model in order to demonstrate the level of agreement?

Response: (1) TUV uses the discrete-ordinates algorithm (DISORT) with 4 streams and calculate the actinic flux spectra with wavelength range of 280-420 nm in 1 nm steps and resolution. I have added it in the manuscript. Aerosol profile is given by Elterman (1968). (2) I have changed global irradiance spectra into actinic flux spectra.

(3) I have simulated actinic flux by TUV to compare with observed results during August 2012, when we have observed SSA data. The agreement between simulation and observation is within 15%. For other time, the simulation couldn't be carried out well due to lack of measured SSA data.

Line 206-210: In order to solve the radiative transfer equation, TUV uses the discrete-ordinates algorithm (DISORT) with 4 streams and calculates actinic flux spectra with wavelength range of 280-420 nm in 1 nm steps and resolution. Measured temperatures were used to calculate the absorption cross sections and quantum yields.
* * *
5) In line 419, the enhanced aerosol level in Bejing is quantified (4-year mean aod = 0.76±0.76). Some references to the studies should be included.

Response: The 4-year mean AOD=0.76±0.75 is calculated by observed AOD during 2012-2015.
* * *
6) In Line 254: "....according to another study in urban Bejing, ..", the reference of the study should be included.

Response: Thank you and I have added the reference.

Line 262-264: According to another study in urban Beijing, the higher the RH, the smaller the slope, and the higher the PBLH, the smaller the slope (Zheng, C. W et al., 2017).
* * *
7) Figure 6: Similar results have been obtained by Bais et al., 2005, Krotkov et al., 2004
and Kazadzis et al., 2017). Is this AOD -SSA dependence from August 2012 obvious during all seasons ? For which wavelength are SSA values given? As both parameters have a wavelength dependence and since PF ozone "effective" wavelengths are ~305-315nm, could this dependence play some role in the provided analysis of the AOD

and

SSA effects on PFs. ?

Response: Thank you, I have added the sentence "Similar results have been obtained by Bais et al., 2005, Krotkov et al., 2005 and Kazadzis et al., 2017)." In the manuscript. We just observed SSA in August 2012 and thus the AOD-SSA dependence is just available in summertime but unavailable in other seasons. SSA values are at 525nm. The following figure is the relationship between AOD and AERONET based SSA (440nm) during 2012-2015 for all seasons. There is a slight positive correlation between AOD and SSA during 2012-2015 for all seasons. We didn't use AERONET based SSA in this study because that: (1) AERONET based SSA have a large uncertainty; (2) There are only 10-20 data of SSA for most months, which is much fewer than AOD data. Sorry, I don't understanding the meaning of PFs.

[Figure]

Figure 1. The relationship between AOD and AERONET based SSA.

Line 343-344: Similar results have been obtained by Bais et al., 2005, Krotkov et al., 2005 and Kazadzis et al., 2012.

- - - - - - - - - - - - - - - - - - - - - - - - - - - - - - - - - - - - - - - - - - - - - - - - - - - - - -

8) Figures 4 and 7: Some commentation on the scatter of J's would be helpful Technical corrections.

Response: Thank you and I have added some comments on the scatter of J versus AOD in page 16 and page 19.

Line 324-325: The scatter of these points is mainly due to variations in ozone column and temperature.

Line 388-389: The scatter of these points is due to the relatively large classification width of SZA to a large extent.
* * *
Line 249-250: Repetition of "in summer" "This implies that the aerosols in summer have stronger extinction capacity in summer than in winter"

Response: Thank you and I have revised it.

Line 257-258: This implies that the aerosols in summer have stronger extinction capacity than in winter.
* * *
Lines 384 &385: cos(SZA) instead of SZA

Response: Thank you and I have revised it.

Line 398-402: The slope of $j(NO_2)$ vs AOD also displays a significant dependence on cos(SZA). The slope increases as cos(SZA) increases from 0 to 0.5 and then decreases as cos(SZA) increases from 0.5 to 1.
* * *
Line 423: "..The result of this study is comparable to the reduction ratio of this study possibly due to..". Probably the one "this study" refers to the previous study mentioned,

Hodzie et el. 2007 and the second one to the authors study, it would be helpful to rephrase.

Response: Thank you and I have revised it. "..The result of Hodzic et al. (2007) is comparable to the reduction ratio of this study possibly due to.."

Line 437-438: The result of Hodzic et al. (2007) is comparable with the reduction ratio of this study possibly due to the equivalent levels of AOD and SSA.
* * *
Line 559: "...in August 2014..", refers to the field campaign in August 2012, mentioned in the paper.

Response: Thank you and I have revised it.

Line 572-573: In order to evaluate the effects of aerosols on ozone production rate, we carried out an observation campaign in August 2012.

**A list of all relevant changes made in the manuscript:**

The relevant manuscript changes are noted in red text.

**Line 27:** Both $j(O^1D)$ and $j(NO_2)$ display significant dependence on AOD (380nm) with a nonlinear negative correlation.

**Line 30-34:** the actinic flux decreases with AOD. The absolute values of slopes are equal to $4.2\text{-}6.9 \cdot 10^{-6}$ s$^{-1}$ and $3.2 \cdot 10^{-3}$ s$^{-1}$ per AOD unit for $j(O^1D)$ and $j(NO_2)$ respectively at SZA of 60° and AOD smaller than 0.7, both of which are larger than those observed in a similar, previous study in the Mediterranean.

**Line 34-36:** This indicates that the aerosols in urban Beijing have a stronger extinction effect on actinic flux than absorptive dust aerosols in the Mediterranean.

**Line 39-42:** According to the parametric equation, aerosols lead to a decrease in seasonal mean $j(NO_2)$ by 24% and 30% for summer and winter, respectively, and the corresponding decrease in seasonal mean $j(O^1D)$ by 27% and 33% respectively, compared to an aerosol-free atmosphere (AOD = 0).

**Line 44-46:** The simulation results shows that the monthly mean daytime net ozone production rate is reduced by up to 25% due to the light extinction of aerosols.

**Line 61-64:** In addition, the photolysis of $NO_2$ produces $O^3P$, and then $O^3P$ reacts with $O_2$ to produce $O_3$, as shown by reactions R3 and R4, which is the only significant chemical source of ozone in the troposphere (Finlayson-Pitts et al., 2000).

**Line 72-77:** Since the photolysis rates are proportional to the actinic flux and not all stations acquire a $2\pi$ spectroradiometer or chemical actinometers for J measurements, several methods have been developed to determine actinic flux and photolysis frequencies from ground based measurements of irradiance (Kylling et al 2003, Kazadzis et al. 2000, 2004, Topaloglou et al. 2005, Trebs et al. 2009).

**Line 85-88:** Previous studies showed that scattering aerosols can enhance the actinic flux throughout the troposphere, while absorptive aerosols reduce the actinic flux throughout the boundary layer (Jacobson, 1998; Dickerson et al., 1997; Castro et al., 2001; Flynn et al., 2010).

**Line 98-100:** Therefore, it is necessary to quantitatively evaluate the effect of aerosols on photolysis frequencies for a better understanding of ozone formation under highly polluted conditions.

The observed data of related influential factors of the photolysis frequencies are taken as the model's input to calculate the photolysis frequencies.

**Line 107:** This method

**Line 132-133:** Previous model studies have shown that aerosols in China can affect ozone production by changing the photolysis frequencies.

**Line 136:** photolysis frequencies

**Line 146-147:** Lou et al (2014) found that with aerosols, annual mean photolysis frequencies.

**Line 157-158:** The relationship between photolysis frequencies and AOD is adequately compared with previous study in the Mediterranean (Casasanta et al., 2011; Gerasopoulos et al., 2012).

**Line 168-171:** From August 2012 to December 2015, $j(O^1D)$ and $j(NO_2)$ were measured continuously at PKUERS site. The data of the period during October 2012 to March 2013 and August 2015 are missed due to instrument maintenance and other measurement campaigns.

**Line 179:** (Shetter and Müller, 1999)

**Line 179-182:** The spectroradiometer consisted of a single monochromator with a fixed grating (CARL ZEISS), an entrance optic with a $2\pi$ steradian (sr) solid angle quartz diffusor and a 2048×64-pixel photodiode array detector.

**Line 183-185:** A 1000 W National Institute of Standard and Technology (NIST) traceable lamp was used for calibration under laboratory conditions (Bohn et al., 2008).

**Line 186-192:** For $j(O^1D)$, the quantum yields used were taken from Matsumi et al.(2002), while the ozone cross section was derived from Daumont et al. (1992) and Malicet et al. (1995). Measured temperatures were used to retrieve ozone absorption cross section and quantum yield. For $j(NO_2)$, the quantum yields used were taken from Bass et al. (1976) and Davenport et al. (1978), while the cross section was derived from Jones and Bayes (1973), Harker et al. (1977) and Davenport (1978).

**Line 192-194:** The calculated photolysis frequencies had a time resolution of 10 s and an accuracy of ±10% including uncertainties associated with the quartz receiver and stray-light effects (Edwards and Monks, 2003).

**Line 195-198:** The optical properties of aerosols were measured by a CIMEL solar photometer (AERONET level 2 data collection, http://aeronet.gsfc.nasa.gov/) and the site selected   is the Beijing-CAMS site (39.93°N, 116.32°E), which is 6.4km from the PKUERS site.

**Line 207-211:** Additionally, at this wavelength we can better compare with the results of Gerasopoulos et al. (2012). The daytime clear-sky conditions were identified according to the presence of AOD data of AERONET since AOD data are unavailable under cloudy conditions. AE were also acquired from ARONET.

**Line 211-212:** The SSA (525nm) data were derived from a field campaign undertaken in August 2012.

**Line 214-216:** Five-minute averages of AOD, SSA, and photolysis frequencies were analyzed in this study.

**Line 216-222:** The total ozone column was obtained by OMI (Ozone Monitoring Instrument) for the year 2012-2015, using overpass data (http://www.temis.nl/protocols/O3global.html) (Henk et al., 2003). In addition, meteorological parameters such as temperature, relative humidity, and pressure were simultaneously observed at this site. Table 1 presents total $O_3$ column, temperature, relative humidity, daytime clear-sky fraction and respective standard deviation for different seasons.

**Line 224-226:** The relevant contents and methods of observation are shown in Table 2.

**Line 231-233:** We use the Tropospheric Ultraviolet and Visible (TUV) radiation model (version 5.3) provided by Sasha Madronich (Madronich, 1993).

**Line 231-233:** In order to solve the radiative transfer equation, TUV uses the discrete-ordinates algorithm (DISORT) with 4 streams and calculates the actinic flux spectra with wavelength range of 280-420 nm in 1 nm steps and resolution. Measured temperatures and ozone column were used to calculate the absorption cross sections and quantum yields.

**Line 237-240:** The key aerosol optical properties including AOD, SSA and AE were input into the model to test the effect of aerosols on photolysis frequencies. AE(380/550nm) is taken from AERONET and the mean value of 1.3 during June 2012 - December 2015 is used in TUV model.

**Line 261-263:** Net ozone production is equal to the reaction rate between peroxy radicals ($RO_2$ and $HO_2$) and NO minus the loss rate of $NO_2$ and $O_3$ as shown in E2, E3, and E4 as derived by Mihelcic et al. (2003).

**Line 272-273:** where $\theta$ is the fraction of $O^1D$ from ozone photolysis that reacts with water vapor. i and j represent the number of species of $RO_2$ and alkenes, respectively.

**Line 278-284:** Compared with AOD, $PM_{2.5}$ is a more common proxy to evaluate the level of particulate matter pollution in spite that AOD is a more closely related parameter of photolysis frequencies. As a result, we attempted to analyze the quantitative relationship between $PM_{2.5}$ and AOD to evaluate the influence of $PM_{2.5}$ on AOD and thus on photolysis frequencies.

**Line 297-299:** According to another study in urban Beijing, the higher the RH, the smaller the slope, and the higher the PBLH, the smaller the slope (Zheng, C. W et al., 2017).

**Line 307-311:** Compared with other cities in North China (Tianjin, Shijiazhuang and Baoding) (Ma et al., 2016), the slope in Beijing for winter is significantly higher. Consequently, using $PM_{2.5}$ to estimate AOD has a large uncertainty due to multiple interference factors.

**Line 315-320:** The diurnal cycles of AOD are shown in Figure 2. AOD displays obvious diurnal variation, with relatively high level at noon and low level at dawn and evening. The diurnal variation of $PM_{2.5}$ is - significantly different from AOD.

Line 329: The observed mean daily maxima of photolysis frequencies.

**Line 336:** under aerosol-free conditions.

**Line 337-342:** Additionally, we know that the temperature is lower in Beijing during the winter compared to conditions in Crete. The measured mean temperature in Beijing during winter is equal to 0.53±4.2 °C (Table 1). When we consider the temperature in Crete is 10 °C higher than in Beijing, the lower $j(O^1D)$ of Beijing than Crete is $5.5 \times 10^{-7}$ s$^{-1}$, which is also not able to compensate the $j(O^1D)$ gap between the two sites during winter.

**Line 364-366:** This relatively large classification width is chosen to make sure that there are enough points to fit the relationship between $j(O^1D)$ and AOD.

**Line 367-368:** The scatter of these points is mainly due to variations in ozone column and temperature.

**Line 377-382:** The observed $j(O^1D)$ was at ozone column of 330-360DU and were scaled to the temperature of 298K. AE(380/550nm) = 1.3, ozone column = 345 and Temperature = 298K were used in TUV model for all simulations. Mean Earth-Sun distance was used in the calculations of TUV model and measured j-values were scaled to the mean Earth-Sun distance.

**Line 390-391:** Similar results in other regions have been obtained by Bais et al., 2005, Krotkov et al., 2005 and Kazadzis et al., 2012).

**Line 436-437:** The scatter of these points is due to the relatively large classification width of SZA to a large extent.

**Line 459-476:** The fitting parametric equations for $j(NO_2)$ is shown in Table 5. For $j(O^1D)$, both of $O_3$ column and temperature affect $j(O^1D)$ significantly. Figure S1 presents the dependence of $j(O^1D)$ on ozone column at low AOD level (AOD<0.3) and SZA of (a) 30°±1° and (b) 60°±1°, respectively. Ozone column ranging from 270 to 400 DU leads to $j(O^1D)$ reducing about 50%. In order to evaluate the impact of temperature on $j(O^1D)$, we calculated the ratio of $j(O^1D)$ at measured temperature to $j(O^1D)$ at temperature = 298K ($j(O^1D)/ j(O^1D)_{T=298K}$) (Figure S2). $j(O^1D)/j(O^1D)_{T=298K}$

varied from 0.82 to 1.03 indicating that temperature changed $j(O^1D)$ by no more than 21%. Therefore, temperature played a minor role in changing $j(O^1D)$ compared with ozone column. As a result, when we fitted the relationship among $j(O^1D)$, AOD and cos(SZA), the effect of ozone column is considered but the effect of temperature is not considered. By fitting the relationship at different ozone classes (classification width=30DU), we found that ozone column increasing by 30DU results in $j(O^1D)$ at a constant SZA and AOD decreasing by 18%. Therefore, the parametric equation for $j(O^1D)$ is transformed into the form E6, which reflects the influence of ozone column. The parameters $a_1$-$a_6$ correspond to ozone column range = 300-330 DU, thus we use 315 DU as the weighted standard of ozone column. The fitting parameters $a_1$-$a_6$ for $j(O^1D)$ is shown in Table 6.

**Line 483-486:** The coefficients of determination of the fitting equations are greater than 0.95 for $j(NO_2)$ and $j(O^1D)$ at a certain $O_3$ column, indicating that both of the photolysis frequencies strongly depended on AOD and cos(SZA),

**Line 493-498:** According to the parametric equations, aerosols lead to a decrease in seasonal mean $j(NO_2)$ by 24% and 30% and a decrease in seasonal mean $j(O^1D)$ by 27% and 33% in summer and winter under clear-sky conditions, respectively, compared to an aerosol-free atmosphere. The decreasing ratio of the photolysis frequencies in winter is higher than in summer mainly due to the higher SZA in winter.

**Line 511-512:** The result of Hodzic et al. (2007) is comparable with the reduction ratio of this study possibly due to the equivalent levels of AOD and SSA.

**Line 518-520:**

**Line 547-549:** Figure 8. Since the decreasing amplitude of the daytime ozone production rate is far larger than that of the daytime ozone loss rate, the mean daytime net production rate of ozone is reduced by 25%.

**Line 635-637:** this result was probably related to a higher proportion of scattering aerosols under high AOD conditions than under low AOD conditions.

**Line 641-645:** According to the parametric equation, aerosols lead to a decrease in seasonal mean $j(NO_2)$ by 24% and 30% for summer and winter, respectively, and the corresponding decrease in seasonal mean $j(O^1D)$ by 27% and 33% respectively, compared to an aerosol-free atmosphere.

**Line 646-647:** In order to evaluate the effects of aerosols on ozone production rate, we carried out an observation campaign in August   2012.

**Line 681-686:**

Bais, A. F., Kazantzidis, A., Kazadzis, S., Balis, D. S., Zerefos, C. S., Meleti, C. Deriving an effective aerosol single scattering albedo from spectral surface UV irradiance measurements, ATMOSPHERIC ENVIRONMENT, 39, 1093-1102, DOI: 10.1016/j.atmosenv.2004.09.080, 2005.

Bass, A. M., Ledford, A, E,, and Laufer, A. H., Extinction coefficients of $NO_2$ and $N_2O_4$, J. Res. Nat. Bureau Standards, 80A, 143-162, 1976.

**Line 711-715:**

Daumont, D., Brion, J., Charbonnier, J., Malicet, J. Ozone UV spectroscopy I: absorption cross-sections at room temperature. Journal of Atmospheric Chemistry 15, 145-155, 1992.

Davenport, J.E. Determination of $NO_2$ photolysis parameters for stratospheric modelling, FAA Report No. FAA-EQ-7-14, 1978.

**Line 723-725:**

Eskes, H. J., Van Velthoven, P. F. J., Valks, P. J. M., Kelder, H. M. Assimilation of GOME total ozone satellite observations in a three-dimensional tracer transport model, Q.J.R.Meteorol.Soc. 129, 1663-1681, doi:10.1256/qj.02.14, 2003.

**Line 778-779:**

Harker, A. B., Ho, W., and Ratto, J. J. Photodissoclation quantum yields of $NO_2$ in the region 375 to 420 nm, Chem Phys. Lett. 50, 394-397, 1977.

**Line 807-828:**

Jones, I. T. N. and Bayes, K.D. Photolysis of nitrogen dioxide, J. Chem. Phys. 59,

4836-4844, 1973.

Kazadzis, S., Bais, A. F., Balis, D., Zerefos, C. S., and Blumthaler, M. Retrieval of down-welling UV actinic flux density spectra from spectral measurements of global and direct solar UV irradiance, J. Geophys. Res., 105, 4857-4864, 2000.

Kazadzis, S., Topaloglou, C., Bais, A. F., Blumthaler, M., Balis, D., Kazantzidis, A., Schallhart, B. Actinic flux and $O^1D$ photolysis frequencies retrieved from spectral measurements of irradiance at Thessaloniki, Greece, ATMOSPHERIC CHEMISTRY AND PHYSICS, 4, 2215-2226, DOI: 10.5194/acp-4-2215-2004, 2004.

Kazadzis, S., Amiridis, V., and Kouremeti, N. The Effect of Aerosol Absorption in Solar UV Radiation, Advances in Meteorology, Climatology and Atmospheric Physics, 1041-1047, 2012.

Krotkov, N., Bhartia, P. K., Herman, J., Slusser, J., Scott, G., Labow, G., Vasilkov, A. P., Eck, T. F., Dubovik, O., Holben, B. N. Aerosol ultraviolet absorption experiment (2000 to 2004), part 2: Absorption optical thickness, refractive index, and single scattering albedo, OPTICAL ENGINEERING, 44, 4, 041005. DOI: 10.1117/1.1886819, 2005.

Kylling, A., Webb, A. R., Bais, A. F., Blumthaler, M., Schmitt, R., Thiel, S., Kazantzidis, A., Kift, R., Misslebeck, M., Schallhart, B., Schreder, J., Topaloglou, C., Kazadzis, S., and Rimmer, J.: Actinic flux determination from measurements of irradiance, J. Geophys. Res., 108 (D16), 4506-4515, 2003.

**Line 875-882:**

[revised manuscript text omitted]

---

## Author Response (AR3)

Corresponding to mshao@pku.edu.cn.

We greatly appreciate the time and efforts that the Referees spent in reviewing our manuscript. The comments are really thoughtful and helpful to improve the quality of our paper. We have addressed each comment below, with the Referee comment in black text, our response in blue text, and relevant manuscript changes noted in red text.
* * *
I think the main difference is that aeronet ssa represent the column and in situ ssa what happens near the ground plus the drying that you mention. I think this have to be clarified.
Response: Many thanks, and I have clarified it. As we used the measured hygroscopic factor (Liu et al., 2009) and measured RH to correct the SSA, the drying didn't contribute to the difference.
* * *
The 11%-16% due to SSA in situ / aeronet differences you are mentioning is very important. It means that due to the in situ SSA use, you are constantly underestimating the PFs by this amount. I think this have to be reported.
Response: Many thanks. I have added this part in the manuscript.
* * *
However, if you have a look at the following paper of Krotkov et al.
https://www.researchgate.net/publication/237741790_Aerosol_ultraviolet_absorption_experiment_2002_to_2004_part_2_Absorption_optical_thickness_refractive_index_and_single_scattering_albedo/figures, and Corr et al.,
https://www.atmos-chem-phys.net/9/5813/2009/
they point out that SSA in the UV can be lower than the visible. So in general this 11-16% could be less.
Response: Many thanks. I have added this part in the manuscript.
* * *
**Line 408-426:**

It worth noting that the mean near-ground SSA (525nm) in August 2012 (0.88±0.08) is significantly lower than the mean AERONET based SSA (440nm) in the same period (0.94±0.02) and in summer (0.94±0.02). The different wavelength plays a minor role in the different SSA according to the wavelength dependence of

AERONET based SSA in the range of 440-1020 nm (Figure S3). This difference is mainly because that the AERONET based SSA represents the column and in situ SSA happens near the ground. The effect of the difference in SSA (0.88 vs 0.94) results in photolysis frequencies changing by 11%-16% according to TUV model. It means that due to the in situ SSA use, the photolysis frequencies tend to be underestimated. Krotkov et al. (2005) and Corr et al. (2009) pointed out that SSA in the UV can be lower than the visible. So in general this 11-16% could be less. The AERONET based SSA generally reproduces well the slope of $j(O^1D)$ versus AOD in spite that it significantly underestimates the absolute value of the slope at low AOD range (AOD<0.7), which is probably due to the uncertainty of AERONET based SSA in low AOD range. In addition to the uncertainty of SSA, both of SSA at 440nm and at 525nm differ from the 305-315nm wavelength range of $j(O^1D)$, which is likely to lead to some uncertainties for the analysis of the relationship between $j(O^1D)$ and AOD.

Line 718-722:

[revised manuscript text omitted]